# Structural basis for Vipp1 membrane binding: from loose coats and carpets to ring and rod assemblies

Benedikt Junglas [1], David Kartte[1,2], Mirka Kutzner[3], Nadja Hellmann [3], Ilona Ritter[1], Dirk Schneider [3,4] & Carsten Sachse [1,2] ✉

Vesicle-inducing protein in plastids 1 (Vipp1) is critical for thylakoid membrane biogenesis and maintenance. Although Vipp1 has recently been identified as a member of the endosomal sorting complexes required for transport III superfamily, it is still unknown how Vipp1 remodels membranes. Here, we present cryo-electron microscopy structures of *Synechocystis* Vipp1 interacting with membranes: seven structures of helical and stacked-ring assemblies at 5–7-Å resolution engulfing membranes and three carpet structures covering lipid vesicles at ~20-Å resolution using subtomogram averaging. By analyzing ten structures of N-terminally truncated Vipp1, we show that helix α0 is essential for membrane tubulation and forms the membrane-anchoring domain of Vipp1. Lastly, using a conformation-restrained Vipp1 mutant, we reduced the structural plasticity of Vipp1 and determined two structures of Vipp1 at 3.0-Å resolution, resolving the molecular details of membrane-anchoring and intersubunit contacts of helix α0. Our data reveal membrane curvature-dependent structural transitions from carpets to rings and rods, some of which are capable of inducing and/or stabilizing high local membrane curvature triggering membrane fusion.

Members of the phage shock protein A (PspA) family have recently been identified as bacterial endosomal sorting complexes required for transport (ESCRT) III proteins by structural and phylogenetic analyses[1–3]. Similar to their eukaryotic counterparts, PspA family proteins are known to form large homo-oligomeric complexes, such as rings or helical rods, and their proposed physiological functions include membrane maintenance and remodeling. A prominent member of the PspA family is the vesicle-inducing protein in plastids 1 (Vipp1), also known as IM30 (inner membrane-associated protein of 30 kDa). Vipp1 is conserved and essential in most oxygenic photoautotrophic organisms and is thought to have been transferred from cyanobacteria to the chloroplasts of algae and plants[4–7]. Vipp1 binds to negatively charged membranes and

membrane binding depends on stored curvature elastic stress[8,9]. Membrane binding has been suggested to be mainly mediated by Vipp1's N terminus, although N-terminally truncated Vipp1 constructs have been shown to be capable of membrane binding[9–11]. Vipp1 appears to have two functions: in membrane maintenance (that is, during heat or light stress) and in thylakoid membrane (TM) biogenesis and/or remodeling[12–18]. The precise structural mechanism underlying both functions and Vipp1's membrane-bound structure are still enigmatic.

The cryo-electron microscopy (cryo-EM) structures of Vipp1 rings in the absence of membranes revealed that the Vipp1 monomer structure contains six α-helices connected by short loops[1,3,19], while a predicted seventh C-terminal α-helix, discriminating Vipp1 from PspA[6,20],

[1]Ernst-Ruska Centre for Microscopy and Spectroscopy with Electrons, ER-C-3/Structural Biology, Forschungszentrum Jülich, Jülich, Germany. [2]Department of Biology, Heinrich Heine University, Düsseldorf, Germany. [3]Department of Chemistry, Biochemistry, Johannes Gutenberg University Mainz, Mainz, Germany. [4]Institute of Molecular Physiology, Johannes Gutenberg University Mainz, Mainz, Germany. ✉e-mail: c.sachse@fz-juelich.de

could not be resolved. Ring structures with variable diameters and rotational symmetry have been determined in the absence of membranes (*Synechocystis C14–C18* (ref. 1) and *Nostoc C11–C17* (ref. 3)). The rings consist of six to seven stacked layers that are formed from monomers arranged in radial symmetry. The conformations of the monomers change from layer to layer, leading to a tapered dome-like shape of the rings[1,3]. The monomers differ at the hinge regions between α3 and α4 (hinge 2) and between α4 and α5 (hinge 3) and in the length of α4. In the rings, a noncanonical nucleotide-binding site has been identified in the layers with the smallest diameters of each ring[1]. Although multiple studies confirmed the nucleoside triphosphatase activity of Vipp1, it does not seem to critically affect the formation of rings, membrane binding and/or membrane fusion[15,21–23].

In addition to rings, Vipp1 forms polymeric assemblies, such as elongated stacked rings, rods or tubes of helical organization and two-dimensional (2D) carpets, although none of these structures have been solved in molecular detail[24–26]. Interestingly, Vipp1 helical assemblies have been shown to engulf lipid membranes in vitro and have also been found to interact with the TM in vivo[1,26]. Vipp1 carpets were observed to form on solid-supported bilayers in vitro and possibly also in vivo during high-light stress[12,14,24]. Thus, rods or tubes and carpets correspond to the predominantly observed Vipp1 structures upon membrane binding. Noteworthily, it has been suggested that Vipp1 carpets are directly involved in membrane protection; upon binding to negatively charged membranes, Vipp1 rings disassemble[8] and transform into membrane-bound carpets that appear to reduce the proton flux across damaged membranes[24]. As carpets have only been observed by low-resolution atomic force microscopy[24], the detailed molecular structure of the carpets is still elusive.

In this study, we analyzed the structure of Vipp1 with membranes using cryo-EM. We observed all previously described species of Vipp1 polymers upon incubation of Vipp1 with membranes. The most common species were helical tubes and stacked rings engulfing membranes and membrane-bound carpets. Among these, we solved the cryo-EM structures of three types of helical tubes and four types of stacked rings at 5–7-Å resolution. In addition to helical tubes or rings, we determined the structures of Vipp1 carpets at three different membrane curvatures to 18–20-Å resolution using subtomogram averaging. Quantitative analysis of membranes with Vipp1 carpets in electron tomograms suggests that increasing Vipp1 coverage on the membrane correlates with higher local membrane curvature up to a point of complete membrane engulfment in small diameter tubes. By analyzing ten additional unique structures of N-terminally truncated Vipp1 in the presence of lipids, we show that helix α0 is a major determinant for the types of polymeric assemblies that are formed by Vipp1 and that helix α0 serves as a membrane anchor for Vipp1. Moreover, by replacing the hinge 2 region of Vipp1 with ten alanines, we generated plasticity-restrained Vipp1 assemblies. The cryo-EM structure of Vipp1 tubes at 3.0-Å resolution revealed the molecular details of the membrane-anchoring helix α0 in addition to conserved Vipp1 intersubunit contacts.

## Results

### Vipp1 forms multiple polymer species upon membrane binding
To investigate the structure of Vipp1 in the presence of bacterial membranes, we refolded Vipp1 in the presence of *Escherichia coli* polar lipid extract (EPL). When refolded in the absence of lipids, Vipp1 forms rings and short stacks of rings (Extended Data Fig. 1a) in agreement with previous EM observations[1,3,25,27,28]. To validate the structural integrity of Vipp1 after refolding, we compared the thermal stability of urea-purified and refolded Vipp1 with natively purified Vipp1, showing that the two samples had essentially identical stabilities (Extended Data Fig. 1b). Qualitative assessments of assembly formation and thermal unfolding supported the notion that refolded protein preparations were structurally similar to the protein purified under native conditions. Upon

refolding of Vipp1 in the presence of EPL, we observed the formation of large irregularly shaped vesicles and even maze-like networks of tubulated membranes in contrast to small circularly shaped liposomes in the absence of Vipp1 (Fig. 1a and Extended Data Fig. 1c). We scrutinized the Vipp1 cryo-EM micrographs and identified isolated rings and elongated rods of stacked rings, tubes and vesicles covered with carpet structures (Fig. 1b, top). The observed structures are heterogeneous; for example, rings have different sizes and the tubular structures are curved and have kinks and variable diameters along the tube axis. When we analyzed the segmented Vipp1 polymer structures by image classification methods, we found 48% being present as elongated evenly indented rods of stacked rings, 45% consisting of two types of tubes with distinct helical lattices and 7% corresponding to regular 2D carpet structures (Supplementary Notes and Fig. 1b, bottom). Taken together, Vipp1 reconstituted with membranes forms multiple polymer species, ranging from rings to extended rods, helical rods, carpets and loose coats.

### Type I and II tubes have different monomer orientations
Next, we set out to analyze the structures of the helical Vipp1 tubes. Using single-particle-based helical reconstruction, we resolved the structures of type I tubes at 5.1-Å resolution and two structures of type II tubes at 7.0- and 7.1-Å resolution (Table 1). We flexibly fitted models according to the typical Vipp1 topology (with the unified nomenclature for the Vipp1 helices[19]) into the determined cryo-EM density maps (Fig. 2a,b). The type I and II tubes differ in their diameter and the arrangement of the monomers relative to the tube axis, as well as the diameter and thickness of the engulfed lipid bilayer tube (Supplementary Notes, Fig. 2b,c and Extended Data Fig. 2a,b). Although the helical tubes display two distinct monomer topologies with respect to the tube–membrane axis, the molecular contacts between the monomers or of the monomers with the membrane remain essentially identical between the different assemblies; α1–α4 contacts stabilize the core, α1–α3 hairpin-to-α5 contacts stabilize the periphery and α0 mediates the interaction with the outer leaflet of the membrane (Fig. 2d–f and Extended Data Fig. 2c,d).

### Joined Vipp1 rings induce high local membrane curvature
In the above-presented micrograph, we identified end-to-end joined rings (that is, stacked rings) with different rotational symmetries (Fig. 1b). On the basis of the cryo-EM images, we determined the three-dimensional (3D) structures of *C11–C14* stacked rings at resolutions of 6.8–6.9 Å (Fig. 3a and Table 1). We flexibly fitted models of the available Vipp1 ring structures[1,3] into our density maps and found that the available models had good overlap with our reconstructions. Thus, stacked Vipp1 rings appear to have the same structure as individual rings, with the monomer structures and hairpin angles relative to the membrane axis changing in each layer from the bottom to the top layer of each ring. Nevertheless, our structures included density for the membrane bilayer. The contact to the bilayer is mediated by α0, as shown for the type I and II tubes (Supplementary Notes, Fig. 3a,b and Extended Data Fig. 3a). The engulfed membrane is most constricted at the position where two rings touch, further increasing the local curvature (Fig. 3c,d and Extended Data Fig. 3b–d).

### Helix α0 is critical for membrane tubulation and affects the Vipp1 polymer structure
The here-determined structures of Vipp1 polymers in the presence of membranes indicated that helix α0 is immersed in the outer lipid leaflet of the bilayer. To obtain more insights into its function, we analyzed the interaction of a truncated Vipp1 lacking α0 (Vipp1 (α1–α6)) with membranes using cryo-EM. We found that, in the absence of α0, Vipp1 forms PspA-like rods that are not capable of membrane tubulation and apparently do not interact with membranes (Supplementary Notes and Extended Data Figs. 4 and 5). Thus, α0 is critical for membrane interaction and tubulation of Vipp1 and in controlling the polymer architecture of Vipp1.

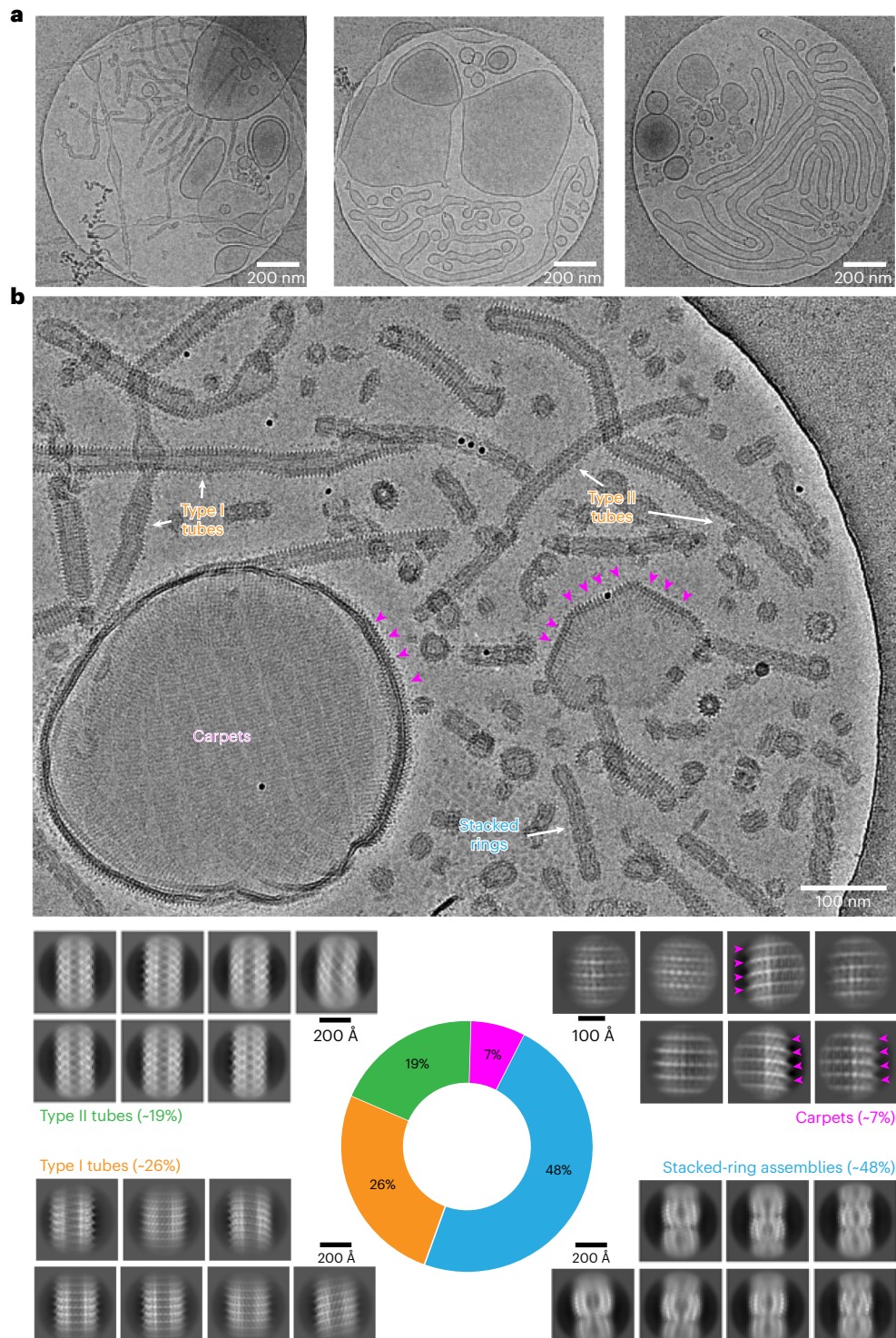

**Fig. 1 | Multitude of different Vipp1 structures after membrane incubation.**
**a**, Three overview cryo-micrographs of Vipp1 after reconstitution with EPL lipids (dataset size: 3,271 micrographs). **b**, Cryo-EM micrograph (dataset size: 3,271 micrographs) and 2D class averages showing a multitude of different Vipp1 structures after reconstitution with lipids. In addition to rings and stacked rings, we identified three additional Vipp1 assemblies: two types of tubular structures with different diameters (type I and type II tubes) and membrane-bound carpet and loose coat structures. Purple arrowheads indicate pronounced spike patterns on Vipp1-coated vesicles.

## Hinge 2 is critical for the plasticity of Vipp1 assemblies

As Vipp1 structures in the presence of lipids reveal a remarkable degree of structural plasticity (Fig. 1b), we set out to create a Vipp1 mutant with restrained flexibility of the Vipp1 monomer (Fig. 5a and Extended Data Fig. 6a). The resulting variant Vipp1 dL10Ala indeed showed reduced structural plasticity when reconstituted with membranes, as it mostly formed type II tubes (69%), including single rings (27%) and stacked rings (3%), and type I tubes (2%) (Fig. 4b and Extended Data Fig. 6b). Therefore, we conclude that the mutated hinge 2 region is a critical element for the structural plasticity of Vipp1 assemblies. While we could reconstruct most of the structures formed by Vipp1 dL10 at low resolution (Supplementary Notes, Table 1 and Extended Data Fig. 6c,d),

**Table 1 | Data collection, image processing and model refinement for Vipp1 assemblies**

| Vipp1+EPL (helical tubes) | | | |
|---|---|---|---|
| | **Type I tubes (305 Å)** | **Type II tubes (290 Å)** | **Type II tubes (275 Å)** |
| Videos | 3,217 | 3,217 | 3,217 |
| Magnification | ×49,000 | ×49,000 | ×49,000 |
| Voltage (kV) | 200 | 200 | 200 |
| Total dose (e⁻ per Å²) | 48 | 48 | 48 |
| Defocus range (µm) | 1.0 to 3.0 | 1.0 to 3.0 | 1.0 to 3.0 |
| Physical pixel size (Å) | 1.737 | 1.737 | 1.737 |
| Detector | Gatan K3 | Gatan K3 | Gatan K3 |
| Symmetry imposed | 1.79-Å rise 124.4° rotation *C1* | 1.96-Å rise 133.5° rotation *C1* | 2.10-Å rise 43.5° rotation *C1* |
| Final no. of segments (ASUs) | 55,189 (1,379,725) | 30,482 (701,086) | 31,649 (664,629) |
| Global map resolution (Å, FSC=0.143) | 5.1 | 7.1 | 7.0 |
| Local map resolution range (Å, FSC=0.5) | 4.9–8.1 | 6.9–9.5 | 6.6–9.0 |
| Initial model used (PDB code) | 7O3W | 7O3W | 7O3W |
| **Model refinement** | | | |
| Model resolution (Å, FSC=0.143) | 5.3 | 6.5 | 7.0 |
| Model *B* factor (Å²) | 147 | 319 | 356 |
| CC mask | 0.74 | 0.76 | 0.57 |
| CC box | 0.26 | 0.21 | 0.56 |
| CC peaks | −0.29 | −0.58 | −0.06 |
| CC volume | 0.75 | 0.74 | 0.53 |
| CC ligands | - | - | - |
| Map sharpening *B* factor (Å²) | −168 | −350 | −100 |
| Model composition | | | |
| Nonhydrogen atoms | 1649 | 860 | 860 |
| Protein residues | 215 | 215 | 215 |
| R.m.s. deviations | | | |
| Bond lengths (Å) | 0.011 | 0.008 | 0.010 |
| Bond angles (°) | 1.829 | 1.621 | 1.620 |
| **Validation** | | | |
| MolProbity score | 1.21 | 0.50 | 0.50 |
| Clashscore | 2.71 | 0.00 | 0.00 |
| Rotamer outliers (%) | 0.6 | 0.0 | 0.0 |
| Ramachandran plot | | | |
| Favored (%) | 97.18 | 99.06 | 100.0 |
| Allowed (%) | 2.82 | 0.94 | 0.0 |
| Disallowed (%) | 0.0 | 0.0 | 0.0 |
| Accession numbers | | | |
| EMDB | EMD-18384 | EMD-18421 | EMD-18420 |
| PDB | 8QFV | 8QHW | 8QHV |

| Vipp1+EPL (stacked rings) | | | | |
|---|---|---|---|---|
| | *C11* | *C12* | *C13* | *C14* |
| Videos | 3,217 | 3,217 | 3,217 | 3,217 |
| Magnification | ×49,000 | ×49,000 | ×49,000 | ×49,000 |
| Voltage (kV) | 200 | 200 | 200 | 200 |
| Total dose (e⁻ per Å²) | 48 | 48 | 48 | 48 |
| Defocus range (µm) | 1.0 to 3.0 | 1.0 to 3.0 | 1.0 to 3.0 | 1.0 to 3.0 |
| Physical pixel size (Å) | 1.737 | 1.737 | 1.737 | 1.737 |

**Table 1 (continued) | Data collection, image processing and model refinement for Vipp1 assemblies**

**Vipp1+EPL (stacked rings)**

| | C11 | C12 | C13 | C14 |
|---|---|---|---|---|
| Detector | Gatan K3 | Gatan K3 | Gatan K3 | Gatan K3 |
| Symmetry imposed | C11 | C12 | C13 | C14 |
| Final no. of segments | 57,470 | 59,515 | 48,541 | 44,233 |
| Global map resolution (Å, FSC=0.143) | 6.9 | 6.9 | 6.8 | 6.9 |
| Local map resolution range (Å, FSC=0.5) | 9.9–13.8 | 8.8–13.6 | 8.8–13.5 | 8.9–14.4 |
| Accession numbers | | | | |
| EMDB | EMD-18422 | EMD-18423 | EMD-18424 | EMD-18425 |

**Vipp1 (α1–α6)+EPL**

| | 280 Å[a] | 300 Å[a] | 320 Å | 340 Å | 360 Å | 370 Å | 390 Å | 400 Å | 405 Å | 420 Å |
|---|---|---|---|---|---|---|---|---|---|---|
| Videos | 5901 | 5901 | 5901 | 5901 | 5901 | 5901 | 5901 | 5901 | 5901 | 5901 |
| Magnification | ×59,000 | ×59,000 | ×59,000 | ×59,000 | ×59,000 | ×59,000 | ×59,000 | ×59,000 | ×59,000 | ×59,000 |
| Voltage (kV) | 300 | 300 | 300 | 300 | 300 | 300 | 300 | 300 | 300 | 300 |
| Total dose (e⁻ per Å²) | 15 | 15 | 15 | 15 | 15 | 15 | 15 | 15 | 15 | 15 |
| Defocus range (μm) | 1.0–2.0 | 1.0–2.0 | 1.0–2.0 | 1.0–2.0 | 1.0–2.0 | 1.0–2.0 | 1.0–2.0 | 1.0–2.0 | 1.0–2.0 | 1.0–2.0 |
| Physical pixel size (Å) | 1.4 | 1.4 | 1.4 | 1.4 | 1.4 | 1.4 | 1.4 | 1.4 | 1.4 | 1.4 |
| Detector | F4 | F4 | F4 | F4 | F4 | F4 | F4 | F4 | F4 | F4 |
| Symmetry imposed | 2.18 Å rise 83.4° rotation C1 | 4.09 Å rise 77.7° rotation C2 | 1.96 Å rise −95.7° rotation C1 | 7.48 Å rise −21.7° rotation C4 | 1.76 Å rise 84.9° rotation C1 | 3.34 Å rise 80.3° rotation C2 | 1.57 Å rise −94.6° rotation C1 | 6.03 Å rise −17.6° rotation C4 | 6.06 Å rise −17.6° rotation C4 | 2.73 Å rise 82.0° rotation C2 |
| Final no. of segments (ASUs) | 26,843 (617,389) | 41,036 (533,468) | 77,480 (1,937,000) | 48,681 (340,767) | 145,546 (4,220,834) | 165,878 (2,488,170) | 187,140 (5,988,480) | 77,818 (622,544) | 66,107 (518,856) | 49,285 (887,130) |
| Global map resolution (Å, FSC=0.143) | 6.9 | 7.0 | 6.3 | 6.7 | 6.6 | 6.7 | 6.6 | 7.0 | 6.7 | 7.8 |
| Local map resolution range (Å, FSC=0.5) | 6.3–12.4 | 6.3–10.9 | 5.6–10.8 | 5.3–10.7 | 5.7–10.6 | 5.8–9.7 | 5.5–9.6 | 5.2–10.4 | 5.7–10.4 | 6.9–11.8 |
| Initial model used (PDB code) | 7O3W | 7O3W | 7O3W | 7O3W | 7O3W | 7O3W | 7O3W | 7O3W | 7O3W | 7O3W |
| **Model refinement** | | | | | | | | | | |
| Model resolution (Å, FSC=0.143) | 6.8 | 7.0 | 6.4 | 6.6 | 7.0 | 7.3 | 6.7 | 7.3 | 6.9 | 8.1 |
| Model B factor (Å²) | 76 | 120 | 247 | 203 | 223 | 332 | 304 | 301 | 290 | 313 |
| CC mask | 0.85 | 0.83 | 0.84 | 0.84 | 0.71 | 0.78 | 0.79 | 0.74 | 0.79 | 0.68 |
| CC box | 0.76 | 0.75 | 0.75 | 0.75 | 0.74 | 0.71 | 0.69 | 0.69 | 0.71 | 0.69 |
| CC peaks | 0.40 | 0.45 | 0.42 | 0.43 | 0.15 | 0.27 | 0.31 | 0.20 | 0.29 | 0.03 |
| CC volume | 0.84 | 0.82 | 0.84 | 0.83 | 0.70 | 0.77 | 0.78 | 0.73 | 0.79 | 0.66 |
| CC ligands | - | - | - | - | - | - | - | - | - | - |
| Map sharpening B factor (Å²) | −450.94 | −543.10 | −200.00 | −200.00 | −200.00 | −200.00 | −200.00 | −200.00 | −200.00 | −200.00 |
| Model composition | | | | | | | | | | |
| Nonhydrogen atoms | 768 | 768 | 768 | 768 | 768 | 768 | 768 | 768 | 768 | 768 |
| Protein residues | 192 | 192 | 192 | 192 | 192 | 192 | 192 | 192 | 192 | 192 |
| R.m.s. deviations | | | | | | | | | | |
| Bond lengths (Å) | 0.009 | 0.009 | 0.009 | 0.002 | 0.002 | 0.006 | 0.001 | 0.010 | 0.010 | 0.009 |
| Bond angles (°) | 1.505 | 1.515 | 1.495 | 0.414 | 0.350 | 0.804 | 1.557 | 1.568 | 1.521 | 1.505 |
| **Validation** | | | | | | | | | | |
| MolProbity score | 0.50 | 0.62 | 0.74 | 1.15 | 0.93 | 1.04 | 0.51 | 0.50 | 0.50 | 0.51 |
| Clashscore | 0.00 | 0.00 | 0.31 | 3.63 | 1.73 | 2.55 | 0.03 | 0.96 | 0.00 | 0.02 |
| Rotamer outliers (%) | 0.00 | 0.00 | 0.00 | 0.00 | 0.00 | 0.00 | 0.00 | 0.00 | 0.00 | 0.00 |
| Ramachandran plot | | | | | | | | | | |
| Favored (%) | 99.47 | 97.37 | 97.37 | 98.42 | 98.95 | 98.95 | 98.95 | 99.47 | 98.95 | 100.0 |
| Allowed (%) | 0.53 | 2.63 | 2.63 | 1.58 | 1.05 | 1.05 | 1.05 | 0.53 | 1.05 | 0.00 |

**Table 1 (continued) | Data collection, image processing and model refinement for Vipp1 assemblies**

**Vipp1 (α1–α6)+EPL**

| | 280 Å[a] | 300 Å[a] | 320 Å | 340 Å | 360 Å | 370 Å | 390 Å | 400 Å | 405 Å | 420 Å |
|---|---|---|---|---|---|---|---|---|---|---|
| Disallowed (%) | 0.00 | 0.00 | 0.00 | 0.00 | 0.00 | 0.00 | 0.00 | 0.00 | 0.00 | 0.00 |
| Accession numbers | | | | | | | | | | |
| EMDB | EMD-18426 | EMD-18427 | EMD-18428 | EMD-18429 | EMD-18430 | EMD-18431 | EMD-18432 | EMD-18433 | EMD-18434 | EMD-18435 |
| PDB | 8QHX | 8QHY | 8QHZ | 8QI0 | 8QI1 | 8QI2 | 8QI3 | 8QI4 | 8QI5 | 8QI6 |

**Vipp1 dL10Ala+EPL (helical tubes)**

| | Type II tubes 250 Å | Type II tubes 270 Å | Type IIb tubes 260 Å | Type IIb tubes 270 Å |
|---|---|---|---|---|
| Videos | 29,548 | 29,548 | 29,548 | 29,548 |
| Magnification | ×63,000 | ×63,000 | ×63,000 | ×63,000 |
| Voltage (kV) | 300 | 300 | 300 | 300 |
| Total dose (e⁻ per Å²) | 60 | 60 | 60 | 60 |
| Defocus range (µm) | 1.75 to 2.75 | 1.75 to 2.75 | 1.75 to 2.75 | 1.75 to 2.75 |
| Physical pixel size (Å) | 1.36 | 1.36 | 1.36 | 1.36 |
| Detector | Gatan K3 | Gatan K3 | Gatan K3 | Gatan K3 |
| Symmetry imposed | 2.53 Å rise 62.40° rotation C1 | 2.13 Å rise 53.15° rotation C1 | 6.54 Å rise −157.00° rotation C3 | 4.04 Å rise −34.45° rotation C2 |
| Final no. of segments (ASUs) | 156,661 (13,133,220) | 80,640 (1,854,720) | 410,963 (3,287,704) | 509,356 (6,112,272) |
| Global map resolution (Å, FSC=0.143) | 5.5 | 6.4 | 3.0 | 3.0 |
| Local map resolution range (Å, FSC=0.5) | 5.7–6.6 | 6.1–7.8 | 3.0–4.1 | 3.0–4.2 |
| Initial model used (PDB code) | 7O3W | 7O3W | 7O3W | 7O3W |
| **Model refinement** | | | | |
| Model resolution (Å, FSC=0.143) | 5.7 | 5.8 | 2.9 | 2.9 |
| Model $B$ factor (Å²) | 315 | 272 | 74 | 64 |
| CC mask | 0.65 | 0.75 | 0.87 | 0.83 |
| CC box | 0.57 | 0.54 | 0.63 | 0.60 |
| CC peaks | 0.06 | 0.10 | 0.47 | 0.41 |
| CC volume | 0.64 | 0.74 | 0.83 | 0.80 |
| CC ligands | - | - | - | |
| Map sharpening $B$ factor (Å²) | −340 | −464 | −157 | −165 |
| Model composition | | | | |
| Nonhydrogen atoms | 51,660 | 51,660 | 98,580 | 98,580 |
| Protein | 12,900 | 12,900 | 12,900 | 12,900 |
| R.m.s. deviations | | | | |
| Bond lengths (Å) | 0.009 | 0.010 | 0.009 | 0.010 |
| Bond angles (°) | 1.490 | 1.499 | 0.779 | 0.828 |
| **Validation** | | | | |
| MolProbity score | 0.50 | 0.50 | 1.08 | 1.08 |
| Clashscore | 0.00 | 0.00 | 2.95 | 2.90 |
| Rotamer outliers (%) | 0.0 | 0.0 | 0.0 | 0.0 |
| Ramachandran plot | | | | |
| Favored (%) | 99.53 | 100 | 99.06 | 99.53 |
| Allowed (%) | 0.47 | 0.0 | 0.94 | 0.47 |
| Disallowed (%) | 0.0 | 0.0 | 0.0 | 0.0 |
| Accession numbers | | | | |
| EMDB | EMD-19863 | EMD-19864 | EMD-19865 | EMD-19866 |
| PDB | 9EOM | 9EON | 9EOO | 9EOP |

**Vipp1 dL10Ala+EPL (rings)**

| | C15 rings | C16 rings | C17 rings | C18 rings | C19 rings | C20 rings |
|---|---|---|---|---|---|---|
| Videos | 29,548 | 29,548 | 29,548 | 29,548 | 29,548 | 29,548 |
| Magnification | ×63,000 | ×63,000 | ×63,000 | ×63,000 | ×63,000 | ×63,000 |

**Table 1 (continued) | Data collection, image processing and model refinement for Vipp1 assemblies**

**Vipp1 dL10Ala+EPL (rings)**

| | *C15* rings | *C16* rings | *C17* rings | *C18* rings | *C19* rings | *C20* rings |
|---|---|---|---|---|---|---|
| Voltage (kV) | 300 | 300 | 300 | 300 | 300 | 300 |
| Total dose (e⁻ per Å²) | 60 | 60 | 60 | 60 | 60 | 60 |
| Defocus range (µm) | 1.75 to 2.75 | 1.75 to 2.75 | 1.75 to 2.75 | 1.75 to 2.75 | 1.75 to 2.75 | 1.75 to 2.75 |
| Physical pixel size (Å) | 1.36 | 1.36 | 1.36 | 1.36 | 1.36 | 1.36 |
| Detector | Gatan K3 | Gatan K3 | Gatan K3 | Gatan K3 | Gatan K3 | Gatan K3 |
| Symmetry imposed | *C15* | *C16* | *C17* | *C18* | *C19* | *C20* |
| Final no. of particles | 33.569 | 56,160 | 73,139 | 56,480 | 33,610 | 19,580 |
| Global map resolution (Å, FSC=0.143) | 7.6 | 7.0 | 7.0 | 7.3 | 7.6 | 10.0 |
| Local map resolution range (Å, FSC=0.5) | 7.0–12.6 | 6.1–10.6 | 6.1–8.7 | 6.5–9.5 | 6.7–9.8 | 9.1–14.1 |
| Accession numbers | | | | | | |
| EMDB | EMD-19899 | EMD-19900 | EMD-19901 | EMD-19902 | EMD-19903 | EMD-19904 |
| PDB | - | - | - | - | - | - |

ᵃFrom the Vipp1 (α1–α6)+EPL dataset. ASUs, asymmetric units; CC, correlation coefficient; r.m.s., root-mean-square; EMDB, EM Data Bank.

we could solve the structure of a subclass of type II tubes at 3.0-Å resolution, presumably because of the reduced flexibility. Outside of the different diameters and helical symmetries, the structures of both type II tubes were similar to the wild-type (WT) type II tubes with the monomers arranged approximately 45° to the tube axis (Table 1, Fig. 4c,d and Extended Data Fig. 7a).

Comparing the monomer structures of the mutant with the WT reveals high similarity in the overall monomer architecture. However, the dL10Ala monomer models had an overall extended length and showed less flexibility in the hinge 2 region, especially in single-ring monomers (Extended Data Fig. 7b), which also explains the observed larger diameters of dL10Ala rings, as smaller rings require stronger bended and less extended monomers. The type II tubes have one continuous approximately 170-Å-long α-helix formed by α2, α3 and α4 presumably stabilized by the introduced deca-alanine stretch (Extended Data Fig. 7c). The monomer structure of the type IIb tubes revealed another unique feature of this assembly, namely the fractured structure helix α5 that is split into two separate helices while still maintaining the canonical ESCRT-III α1–α3 hairpin-to-α5 contacts (Extended Data Fig. 7d). The built type IIb model also revealed the detailed structure of hinge 2 in Vipp1 dL10Ala (Extended Data Fig. 7e). As predicted, the flexible loop was replaced by a small kink at A163 and, accordingly, α3 and α4 were elongated, making hinge 2 less flexible. Similarly, the intermolecular and intramolecular interactions within the tube were resolved at near-atomic detail. The intramolecular interaction between helix α1 and α2 is primarily based on hydrophobic interactions (for example, L108, Y104, L97, T65, L61, T57 and I51) (Extended Data Fig. 7f). As shown previously, a critical intermolecular interaction to stabilize higher-order Vipp1 assemblies involves the conserved ₁₆₈FERM₁₇₁ motif[8,24,25]. F168 interacts with a hydrophobic groove (L29, V33, A132, K135 and L139) between helix α1 and α3 of an adjacent monomer (Extended Data Fig. 7g). E169 is directed into a gap between four neighboring monomers and has no direct interaction partners. R170 and M171 point to the lumen of the tubes and interact with helix α0 of two other subunits (R6 (subunit a) and E23 and P25 (subunit b)). Another conserved residue close to the ₁₆₈FERM₁₇₁ motif is E179, interacting with R142 in α3 of an adjacent subunit.

As shown in this study (Extended Data Figs. 4 and 5), α0 was found to be critical for membrane tubulation, while it also affects the Vipp1 polymer structure. Given its key position at the inner wall of the Vipp1 assembly, it forms polar interactions with the adjacent α0 and α4, which may explain the central importance of helix α0 for enabling different assembly states of Vipp1 (Fig. 4e and Extended Data Fig. 7b–d). We also found that the side chains of L3, F4, L7, V10, V11, L15, L18 and V19 making up the hydrophobic face of α0 are directed into the inner tube lumen. They interface with additional density, supporting the notion of interacting with the hydrophobic acyl chains of lipids, respectively. The opposite face of helix α0, directed toward the Vipp1 wall pointing outward of the assembly, is lined by positively charged residues R6, R9 and R12 interfacing with negatively charged residues (for example, D35 (helix α1)), while they could possibly also interact with negatively charged lipid head groups (Fig. 4e, right, and Extended Data Fig. 7h). To experimentally test which α0 residues are involved in membrane interaction, we made use of the observation that the isolated Vipp1 α0 forms an α-helix only when binding to negatively charged membrane surfaces[10]. Thereby, we measured the membrane-binding capacity of different variants of the α0 peptide using circular dichroism (CD) spectroscopy. Compared to the WT peptide, peptides that had R6 and R9 or F4 and V11 replaced by two alanines reduced the membrane affinity to a small extent as they reached similar ellipticity plateaus, while the WT peptide-binding curve decayed faster than the R6A;R9A and F4A;V11A binding curves (Extended Data Fig. 7i). By replacing the eight hydrophobic residues with alanines, thus destroying the amphipathic character of the peptide, we abolished membrane binding completely. This observation indicates that the amphipathic character is essential for membrane interaction while the hydrophobic face appears to be more critical than the positively charged residues. In conclusion, we could show that hinge 2 is a critical element for the structural plasticity of Vipp1 assemblies. Using the plasticity-reduced Vipp1 dL10Ala variant, we successfully determined the structure of Vipp1 tubes at near-atomic resolution and revealed critical interactions of α0 important for the membrane–lipid interaction and for the intermolecular assembly contacts.

**Vipp1 membrane coverage determines membrane curvature**

In addition to helical tubes, stacked rings and rings, we also found carpets and loose coats of Vipp1 on membranes in the cryo-EM micrographs of the Vipp1 lipid sample (Fig. 1b). When taking a closer look at the small Vipp1 coats, we found discontinuous densities on the membrane of vesicles without changing the local curvature (Fig. 5a, white arrowheads). However, when patches exceeded a critical size, they were found on positive local curvature on the vesicular surface introducing noticeable bulges emanating from the membrane plane (Fig. 5a, cyan arrowheads). When membrane-attached ring or stacked

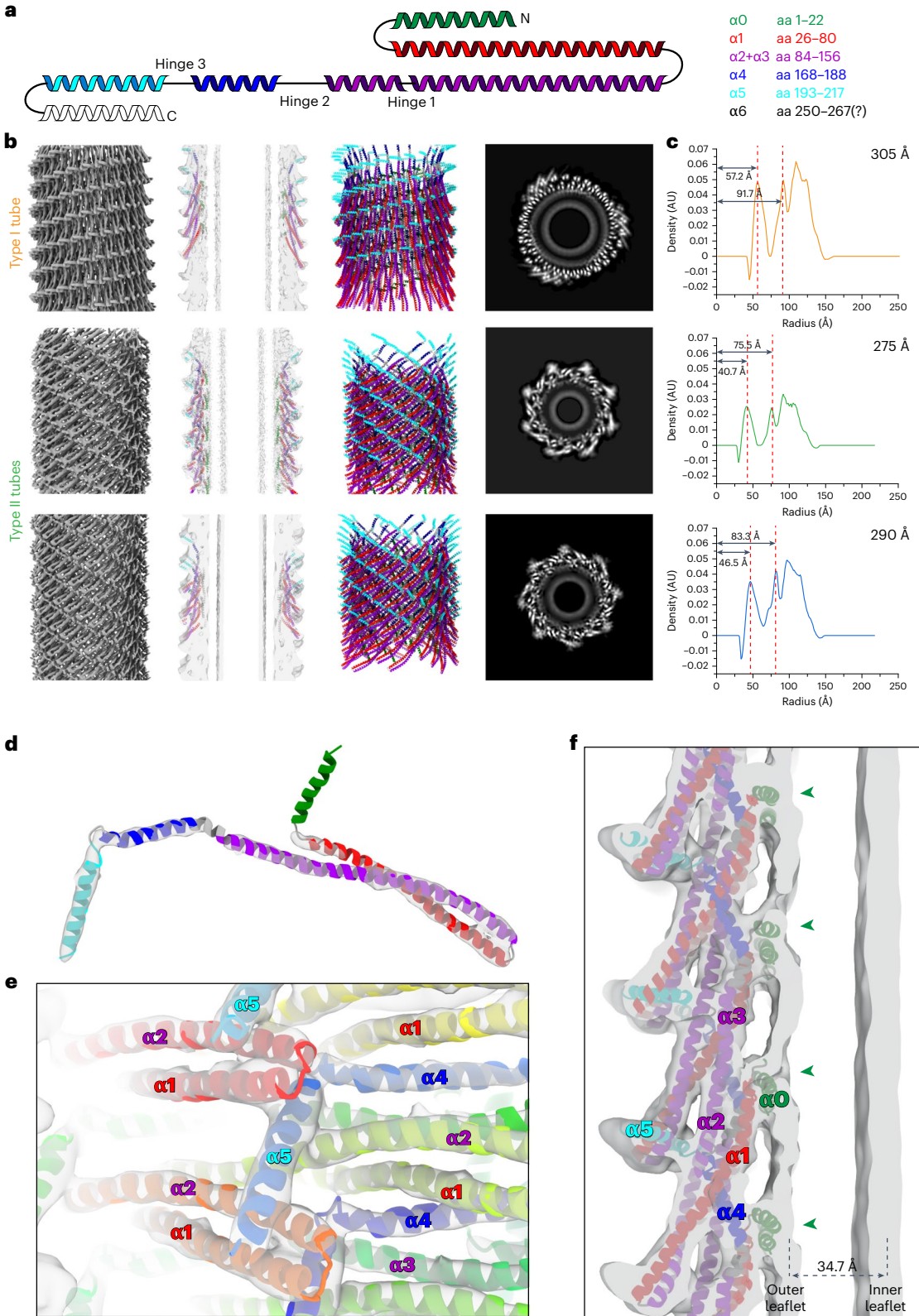

**Fig. 2 | Structural details of type I and type II helical Vipp1 assemblies.**
**a**, Topology of the Vipp1 structure with color-coded helices: α0, green; α1, red; α2 + α3, violet; α4, blue; α5, cyan; α6, white. **b**, Cryo-EM structures of type I (top row) and two type II assemblies (center and bottom row). Left, cryo-EM maps of Vipp1 tubes. Center left, central *xy* slices of cryo-EM maps with the fitted models. Center right, atomic models of Vipp1 tubes in ribbon representation. Right, central *z* slices of cryo-EM maps in grayscale. **c**, Radial density profiles of respective Vipp1 tubes (respective outer diameters of the tubes are displayed in the upper right corner of each plot). Dashed lines indicate the peaks of the inner and outer leaflet densities of the tubulated bilayer. AU, arbitrary units. **d**, Segmented density and modeled atomic Vipp1 monomer structure found in type I tubes showing the ESCRT-III fold. **e**, Density of the Vipp1 type I tubes with the built polymer model showing the polymer core with helices α1–α4, the tip of the α1–α3 hairpin and α5 contact sites. **f**, Vipp1 type I tubes: enlarged view of both leaflets of the tubulated bilayer with the fitted model. Green arrowheads pointing to indentations in the outer leaflet caused by helix α0 interacting with the lipids.

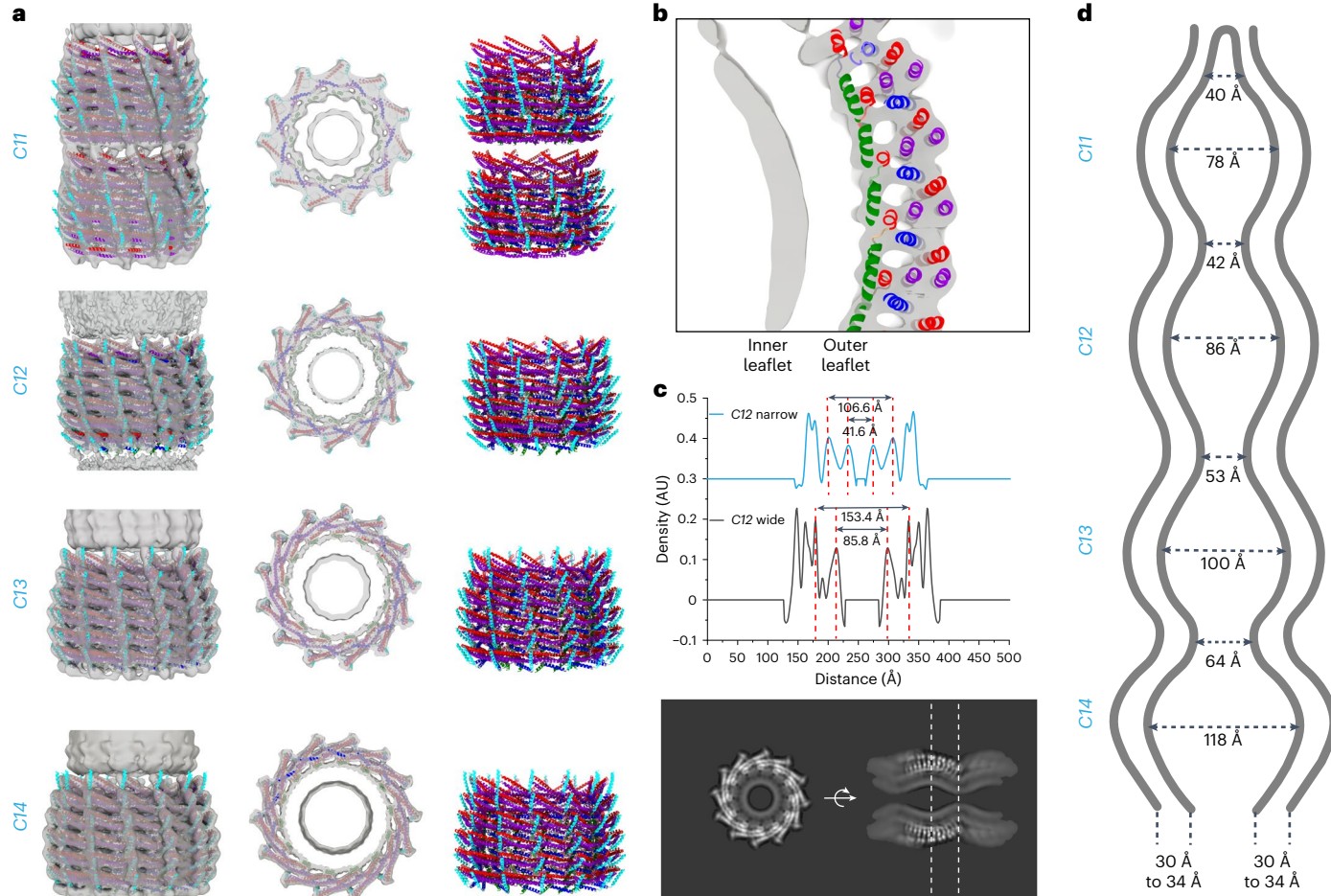

**Fig. 3 | Cryo-EM structures of stacked Vipp1 rings with engulfed membranes.** **a**, Structures of stacked Vipp1 ring assembly with *C11*, *C12*, *C13* and *C14* rotational symmetry (top to bottom rows, respectively). Left, transparent cryo-EM maps with the fitted models. Center, central *z* slices of cryo-EM maps with fitted models. Right, atomic models of Vipp1 rings in ribbon representation (α0, green; α1, red; α2 + α3, violet; α4, blue; α5, cyan). **b**, Enlarged view of a Vipp1 ring in a stacked-ring assembly showing both leaflets of the tubulated bilayer with the fitted model. **c**, Top, density profiles across the stacked *C12* ring. Bottom, *z* (left) and *xy* (right) slices of the map in grayscale, including the locations of the density sections above (dashed lines). **d**, Schematic sketch of the measured membrane distances in stacked Vipp1 rings for the narrowest and widest part of *C11*−*C14* rings and the observed bilayer thicknesses (shown in detail in Extended Data Fig. 3b−d).

rings were observed, they also showed local curvature formation at the host membrane (Fig. 5a, orange arrowheads). In many cases, the curved membrane extended to a small neck connected to the rings or longer stacked rings. Other discernible species of membrane-bound Vipp1 assemblies were carpets that fully covered vesicles (Fig. 5a, red circle). These vesicles typically had a smooth spherical shape without bumps or bulges while they exhibited a pronounced spike pattern at the edges and a regular stripe pattern in the center. A comparison of the cylindrical projection of the Vipp1 type I tubes and 2D class averages of the carpet and their corresponding Fourier transforms revealed that the distances occurring in this regular stripe and spike pattern were very similar, indicating common underlying structures (Extended Data Fig. 8a,b). These distances of the cylindrical projection in type I tubes correspond to the stacking distance of the α1−α2 helical hairpin array, suggesting that a similar hairpin array is also present in Vipp1 carpets.

We further elucidated the 3D structures of Vipp1 on vesicles using segmented tomograms. The segmented tomograms revealed tape-like Vipp1 ribbons that were not membrane attached and that Vipp1 assemblies differ depending on the size of the vesicles (that is, their curvature) they are attached to (Supplementary Notes, Fig. 5b,c and Extended Data Fig. 8c−e). For a more quantitative analysis of the observed Vipp1 membrane assemblies, we determined the membrane occupancy by Vipp1 in relation to the membrane curvature. The membrane occupancy

increased linearly with increasing membrane curvedness between curvedness values of 0.02−0.06. For high-curvature regions (that is, for curvednesses larger than 0.06), the graph indicated a saturation of Vipp1 membrane binding (Fig. 6a and Extended Data Fig. 9a,b). To support the assumption that Vipp1's affinity is higher for liposomes with high curvature, we compared binding of the protein to liposomes of different sizes and confirmed that Vipp1 preferably binds to small liposomes with high membrane curvature (Fig. 6b and Extended Data Fig. 9c). Using subtomogram averaging, we were able to determine structures of Vipp1 carpets for three distinct curvatures (Table 2, Fig. 6c and Extended Data Fig. 9d,e). The carpet structures were remarkably similar to Vipp1 rings with respect to spike separation distance and monomer stacking distance and displayed the characteristic features of all Vipp1 structures determined to date (that is, the typical α1−α4 contacts in the core, α1−α3 hairpin-to-α5 contacts in the periphery and α0-mediated membrane contacts). Lastly, our observations indicated a strong correlation between membrane curvature and the degree of Vipp1 membrane coverage.

## Discussion

In this study, we elucidated the structures of different Vipp1 assemblies bound to membranes from highly ordered symmetrical assemblies to more loosely organized membrane-covering carpets (Fig. 1). The

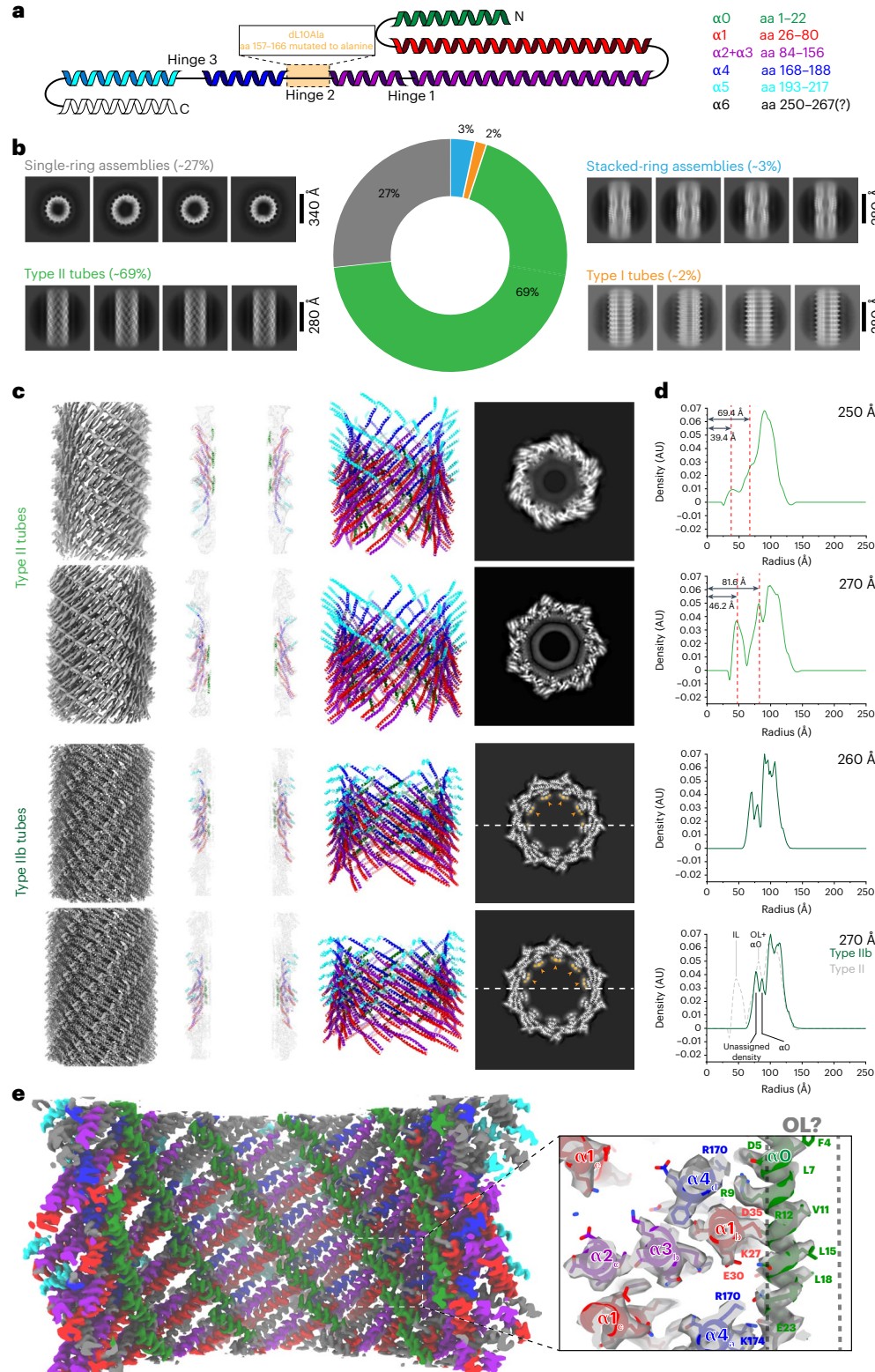

**Fig. 4 | Cryo-EM structures of the plasticity-reduced Vipp1 dL10Ala.**
**a**, Topology of the Vipp1 structure with indicated mutation site for Vipp1 dL10Ala and color-coded helices: α0, green; α1, red; α2 + α3, violet; α4, blue; α5, cyan; α6, white. **b**, The 2D class averages showing different Vipp1 dL10Ala structures after reconstitution with lipids. In addition to rings and stacked rings, we identified type I and type II tubes. **c**, Cryo-EM structures of two type II (top rows) and two type IIb assemblies (bottom rows). Left, cryo-EM maps of Vipp1 dL10Ala tubes. Center left, central *xy* slices of cryo-EM maps with the fitted models. Center right, atomic models of Vipp1 tubes in ribbon representation. Right, central *z* slices of cryo-EM maps in grayscale with highlighted unassigned density in orange.

**d**, Radial density profiles of respective Vipp1 dL10Ala tubes (respective outer diameters of the tubes are displayed in the upper right corner of each plot). Dashed lines indicate the peaks of the inner and outer leaflet densities of the tubulated bilayer. Bottom, profile showing the superposition of type IIb (solid green) and type II (gray dashed) tubes of 270-Å diameter assigned with putative structural elements. **e**, Cryo-EM map of Vipp1 dL10Ala type IIb tubes with surface colored Vipp1 α-helices and enlarged view of the cryo-EM map with the fitted model showing the helix α0 interactions and putative position of a membrane outer leaflet (OL).

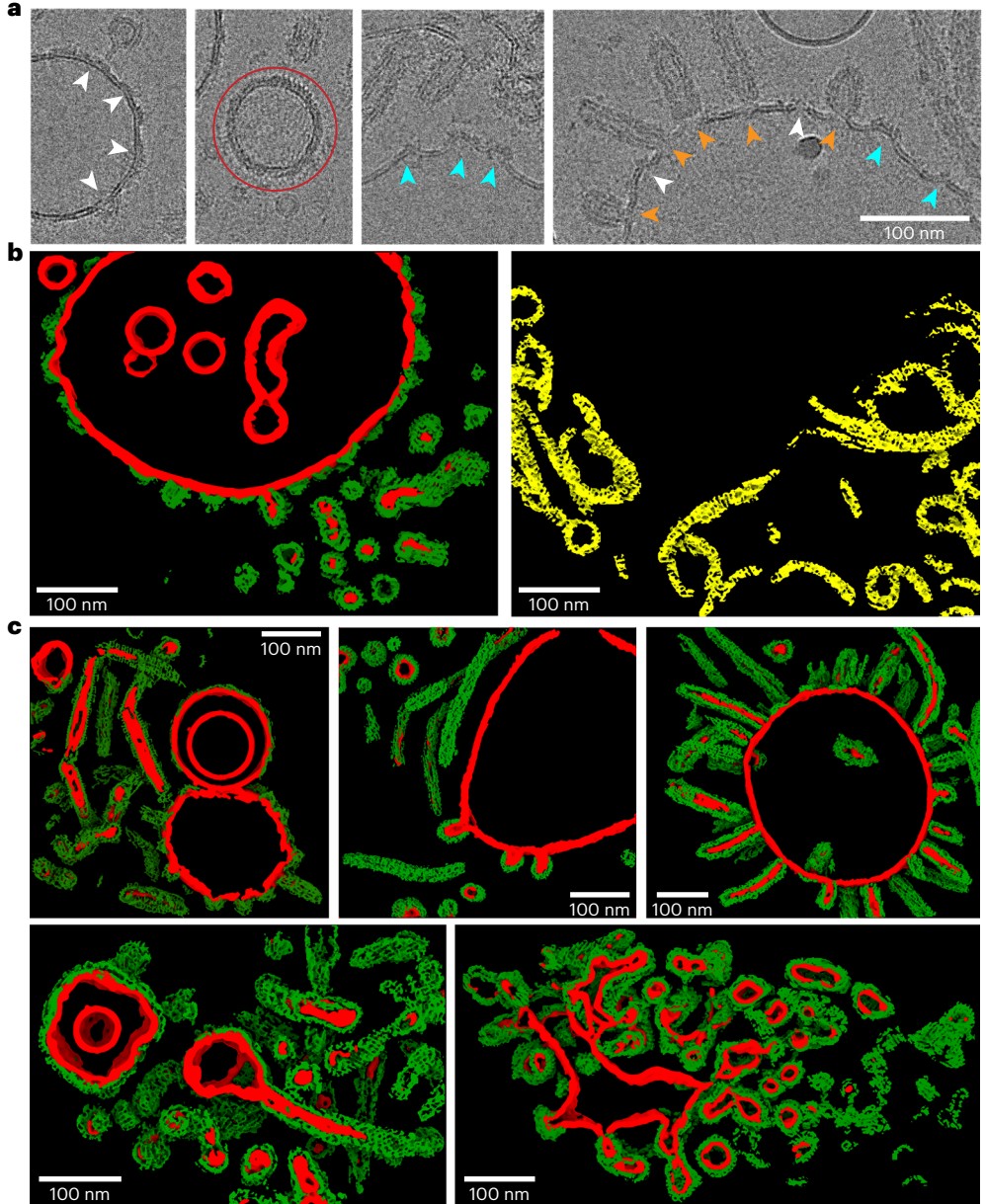

**Fig. 5 | Assemblies of Vipp1 loose coats and carpets on membranes.**
**a**, Cryo-EM micrographs of Vipp1 after lipid reconstitution with EPL (dataset size: 3,271 micrographs). White arrowheads indicate loose coats of Vipp1 on flat membranes. Cyan arrowheads indicate sites of small Vipp1 patches with induced bulges at the membranes. Orange arrowheads indicate sites where Vipp1 ring or rod assemblies bind to the membrane or bud off the membrane. The red circle indicates a vesicle fully covered with a Vipp1 carpet. **b**, The 3D renderings of the same tomogram after segmentation using a progressively trained U-Net. The U-Net was trained to segment membranes, membrane coverings and spirals into separate classes. For better visibility, membranes, Vipp1 and Vipp1 spiraling ribbons were rendered in red, green (left) and yellow (right), respectively. The raw tomogram is shown in Extended Data Fig. 8c. Red, membrane; green, Vipp1 rings, rods and carpets; yellow, Vipp1 spirals. **c**, Gallery of segmented Vipp1 lipid tomograms.

best-organized states were helical tubes that occurred in two different architectural types and stacked-ring Vipp1 assemblies of cyclical symmetries, all of which included lipid bilayer density in the lumen (Figs. 2 and 3). Membrane tubulation of Vipp1 is largely mediated by α0, as truncated Vipp1 (α1–α6) assembled into helical tubes that did not internalize any lipid membrane in the lumen (Extended Data Figs. 4 and 5). By generating the plasticity-restrained Vipp1 dL10Ala mutant, we could narrow down the variability of Vipp1 assemblies and solve the structure of two Vipp1 tubes at 3.0-Å resolution. These structures revealed that one positively charged face of α0 serves as a central contact point to hold α0, α1 and α4 from four adjacent subunits in place while the hydrophobic face can interact with the acyl chains of

engulfed membrane lipids (Fig. 4). Less well-ordered Vipp1 assemblies included membrane-free spiraling ribbons and membrane-associated loose coats, patches and carpets (Fig. 5). Membrane coverage of Vipp1 correlated with higher membrane curvature, leading to the observed more ordered symmetrical Vipp1 complexes (Fig. 6).

Our Vipp1-containing membrane preparations revealed a remarkable diversity of membrane-bound Vipp1 structures suitable for structure determination, in contrast to past studies that worked with preformed rings incubated with membranes[3,8,13,24,26]. However, it is likely that preformed Vipp1 rings are less reactive (Supplementary Notes). Thus, we purified Vipp1 under denaturing conditions and refolded it in the presence of lipids. We identified refolding in the presence of

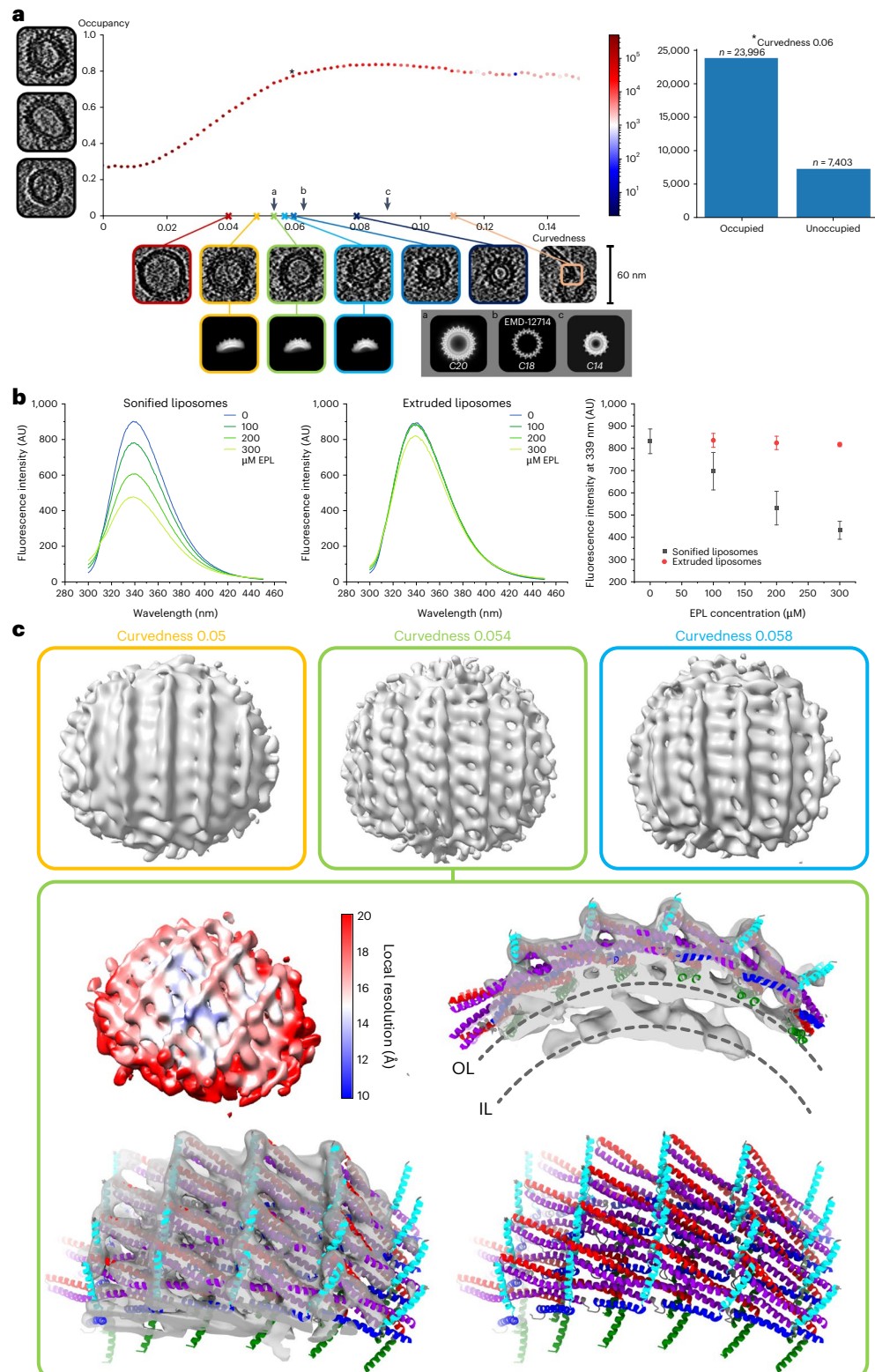

**Fig. 6 | Vipp1 membrane curvature analyses and corresponding subtomogram averages. a**, Membrane curvature in relation to occupancy representing the Vipp1 membrane coverage. Left, example images of different occupancies with fully covered, partly covered and uncovered vesicles. Bottom, example images of vesicles and other shapes with specified curvedness. The 2D averages of subtomogram averages corresponding to the indicated curvedness. Bottom right, the 2D average of *C14*, *C18* and *C20* rings as a comparison. Right, exemplary examination for all segmentation points with a curvedness of 0.06.

**b**, Tryptophan fluorescence spectroscopy of Vipp1 in the presence of sonified (~55-nm diameter) or extruded (~144-nm diameter) liposomes. Solely upon the interaction of refolded Vipp1 with sonified liposomes, a pronounced decrease in quantum yield is observed for small liposomes. Error bars represent the s.e.m. (*n* = 4 biological replicates). **c**, Top, subtomogram averages for curvedness 0.05, 0.054 and 0.058. Left, local resolution map yielding an average resolution of 18 Å. Density maps with fitted atomic model based on the *C20* symmetrization (adapted from PDB 7O3Z). IL, inner leaflet.

**Table 2 | Data collection and image processing of Vipp1 assemblies with subtomogram averaging**

| Subtomogram averaging | | | |
|---|---|---|---|
| Magnification | ×53,000 | ×53,000 | ×53,000 |
| Voltage (kV) | 300 | 300 | 300 |
| Total dose (e⁻ per Å²) | 152 | 152 | 152 |
| Dose per tilt (e⁻ per Å²) | 3.7 | 3.7 | 3.7 |
| Energy filter slit width (eV) | 20 | 20 | 20 |
| Defocus | 3 µm | 3 µm | 3 µm |
| Physical pixel size (Å) | 1.7 | 1.7 | 1.7 |
| Detector | Gatan K3 | Gatan K3 | Gatan K3 |
| Acquisition scheme | Dose-symmetric | Dose-symmetric | Dose-symmetric |
| Tilt range | −60° to +60° | −60° to +60° | −60° to +60° |
| Tomogram number | 123 | 123 | 123 |
| Initial subtomograms | 526,864 | 526,864 | 526,864 |
| Curvedness | 0.05 | 0.054 | 0.058 |
| Final subtomograms | 3,548 | 7,349 | 6,544 |
| Global map resolution (Å, FSC=0.143) | 20.4 | 18.1 | 19.5 |
| Local map resolution range (Å, FSC=0.143) | 20–24 | 12–24 | 18–25 |
| EMDB | – | EMD-18620 | – |

membranes as a condition more capable of membrane remodeling, giving rise to the observed diversity of Vipp1 membrane assemblies while keeping their native ESCRT-III fold intact.

The here-determined Vipp1 stacked rings closely resemble previously observed membrane-free rings[1,3], except for an internalized continuous membrane tube connecting the rings. The rings are built by six unique layers of different monomer structures. The monomers differ at the hinge 2 and 3 regions between α3 and α4, between α4 and α5 and along the length of α4. The angle of the α1−α3 hairpin changes from the bottom to the top layers (Fig. 3). At the bottom layer, the hairpin has a nearly 90° angle to the ring axis; approaching the top layer, the angle decreases to ~60–70°. This effect is more pronounced at smaller ring diameters[1,3]. In contrast to previous studies based on bulk fluorescence resonance energy transfer and fluorescence measurements[29], our stacked-ring assembly structures now clearly show that Vipp1 rings are joined in a polar head-to-tail fashion (Fig. 3a). This head-to-tail interaction is mediated by the docking of the α1−α3 hairpin of the lower ring to α5 of the upper ring. This interaction mode is in good agreement with the observation that substitution of conserved residues at the tip of the α1−α3 hairpin prevents Vipp1 ring stacking[25]. Interestingly, it has been shown that the addition of Mg²⁺ can induce ring stacking in the absence of membranes[25,29], suggesting that Mg²⁺ may stabilize the α1−α3 hairpin-to-α5 interaction. The membrane engulfed in stacked rings exhibits a notable feature as it does not form a uniform tube diameter throughout the stacked-ring assembly. Instead, the membrane follows the tapered wall of the ring and is constricted at the sites where two rings are touching and at the 'open' ends of the stacked rings. Consequently, the membrane undergoes a soft kink of approximately 120° (Fig. 3c,d). The high local curvature renders the membrane suitable for spontaneous fusion with other membranes, particularly at the 'open' ends. In this way, one ring may connect to another membrane-filled ring and thereby elongates the stacked-ring assembly or fuse with a free membrane. In fact, the membrane outline inside stacked rings closely resembles the typical hourglass shape of two fusing membranes that are caged in that highly reactive state by the Vipp1 lattice. The observed membrane curvature changes within the structures are formed by the protein cage and could also result in the local accumulation of a specific lipid subpopulation and redistribution of the used EPL mixtures. Together, our data support the concept that the protein assembly structure stabilizes possible membrane transition states highly suitable for Vipp1-mediated membrane fusion, involving the induction of high local membrane curvature within the Vipp1 rings[3,13,15,25]. The here-determined two types of helical rods represent unique Vipp1 structures. In contrast to the Vipp1 rings, the monomer orientation of the α1−α3 hairpin with respect to the tube axis is the same over the whole length of the tubes. Nevertheless, the monomer orientation among the tubes differs; in type I tubes, the α1−α3 hairpins are oriented parallel to the tube axis, while, in type II tubes, the hairpins are tilted approximately 45° relative to the tube axis. Within the Vipp1 rings, the α1−α3 hairpin orientation changes gradually from approximately 90° at the bottom to approximately 60–70° at the top. Remarkably, from rings over type II tubes to type I tubes, the α1−α3 hairpins accommodate a wide range of orientations along the tubulated membrane axis from perpendicular to parallel (from 90° to 0°) while essentially maintaining the same intermolecular contacts to stabilize the assembly (Fig. 7a). Interestingly, in our plasticity-restrained mutant, the structural diversity of membrane-interacting assemblies was limited to predominantly type II tubes (that is, to hairpins aligned 45° relative to the tube axis). Therefore, it can be expected that the intrinsic structural plasticity and conformational flexibility of WT Vipp1 is required to sample different conformations and transition states that assist in membrane remodeling. This kind of structural orientation change appears to be a unique feature of Vipp1, as the other ESCRT-III proteins with known structures on membrane tubes (that is, SynPspA, CHMP2A + CHMP3, CHMP1B and CHMP1B + IST1) have a fixed orientation relative to the membrane axis[2,30,31] (Extended Data Fig. 9f).

The determined structures of stacked rings and helical tubes revealed an internalized lipid bilayer. The bilayer thickness in all three Vipp1 tubes remained nearly identical at 35 Å, although the inner leaflet radius changed with the tube diameter. This observation differs from the previous reports about CHMP1B and CHMP1B + IST1 (co)polymers where the bilayer thickness decreased with decreasing tube diameter[31]. The bilayer thickness in the Vipp1 stacked rings, however, changed only slightly with the increasing diameter of the rings. This observation may be a result of the high local membrane curvature within the stacked rings. Similar to other ESCRT-III proteins, the membrane contact in the Vipp1 tubes and rings is mediated either by an additional α-helix (that is, α0) in the case of Vipp1 and SynPspA or a short unstructured extension

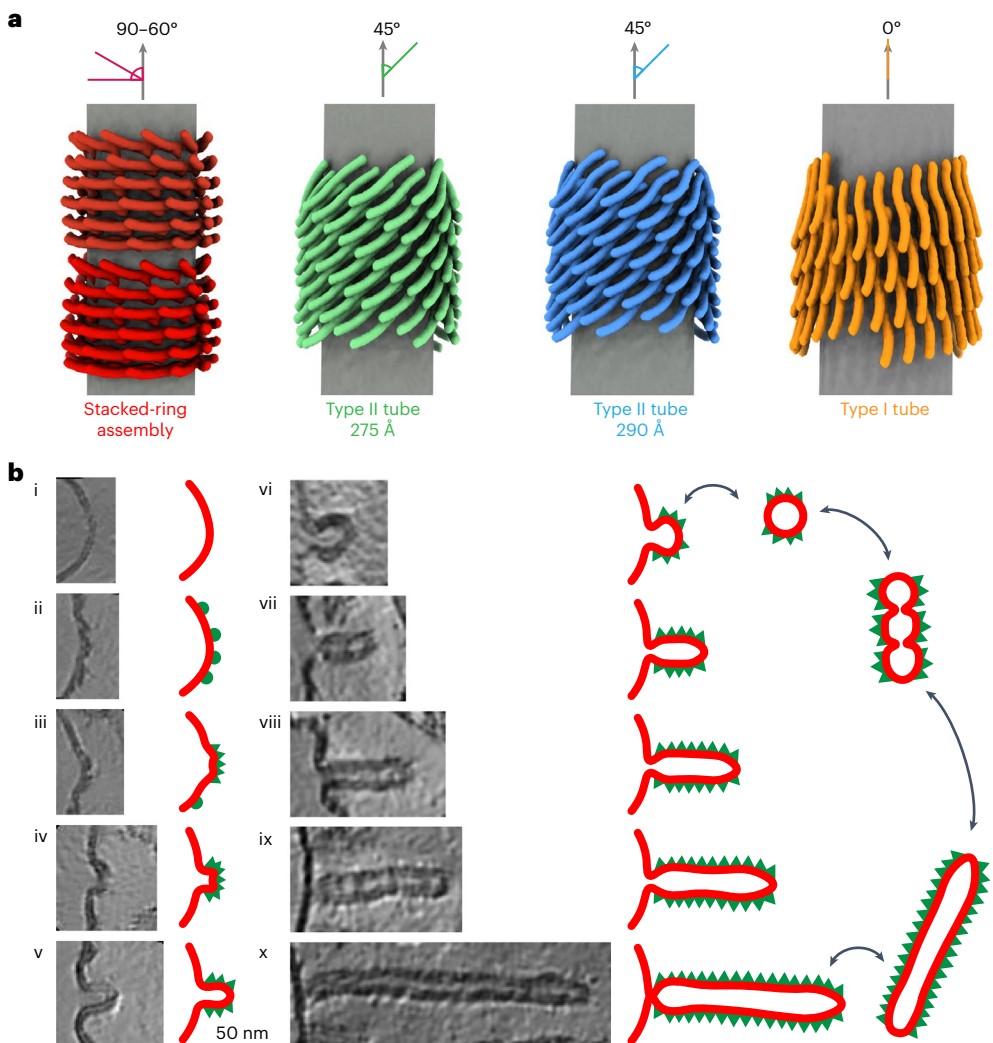

**Fig. 7 | Orientation of the Vipp1 hairpin and sorted tomographic slices of Vipp1 membrane interactions. a**, Schematic view of Vipp1 α1–α3 hairpins relative to the membrane axis in stacked-ring assembly, type II tubes and type I tubes. **b**, The identified structures are sorted with regard to Vipp1 membrane interactions and the degree of membrane deformation. Initially, Vipp1 binds to free membranes (i). When the Vipp1 monomers reach a critical local concentration on the membrane surface (ii), they start to oligomerize and form a membrane bulge (iii). Vipp1 induces membrane curvature upon oligomerization (iv). The induced curvature can result in the formation of a budding Vipp1 oligomer with enclosed membrane (v). This assembly can either bud off the membrane to produce rings (vi) or grows further to extended rod structures (vii–x). The rods can reside at the host membrane or bud off to produce Vipp1-coated membrane tubes. The budded rings can fuse to form stacked rings that finally relax to Vipp1-coated membrane tubes. Please note that this model is limited by the observation of Vipp1 assemblies in micrographs and sorting them into logical sequence.

of helix α1 in the case of CHM1B or CHMP2A + CHMP3 (refs. 2,30–32). However, in contrast to *Syn*PspA helix α0 that protrudes toward the bilayer[2], Vipp1 α0 lies flat in the membrane plane and appears to be partially submerged into the outer bilayer leaflet. The 3.0-Å-resolution structure of a plasticity-restrained Vipp1 dL10Ala mutant revealed the amphipathic positioning of helix α0 with the hydrophobic face directed into the tube lumen to potentially interact with hydrophobic acyl chains while the positive helix face interacts with the inner Vipp1 ring or tube surface and may be capable of interacting with lipid head groups.

Interestingly, truncation of α0 prevents the internalization of membranes into the lumen of the assemblies and/or formation of assemblies on the membrane. This observation is in line with the reports that the isolated α0 of Vipp1 binds to membranes[9,10]. Although it has been reported that truncated versions of Vipp1 lacking α0 are still capable of membrane binding[11], our micrographs and cryo-EM maps show that Vipp1 (α1–α6) does not engulf membranes in the rod lumen and does not assemble on membranes, while it may still be capable of binding lipids in other conditions. Furthermore, substitution

of the conserved residues F4 and V11 in the hydrophobic face of α0 led to severely decreased light resistance and TM swelling of *Synechocystis* cells[1], possibly because Vipp1 could no longer properly interact with membranes. Thus, we conclude that helix α0 is the membrane anchor of Vipp1, in line with the suggested N-terminal ANCHR motif of some eukaryotic ESCRT-III proteins (Snf7, CHMP2B, CHMP3, CHMP2A and CHMP1B)[30–34].

Moreover, the truncation of helix α0 also affects the assembly state of Vipp1. Vipp1 prefers to form rods over rings when residues conserved in helix α0 are substituted[1] or exclusively forms rods when helix α0 is completely absent (this study and a previous study[11]). Additionally, the monomer orientation and rod symmetry differ from the rods observed upon incubation of the full-length protein with membranes while the Vipp1 (α1–α6) rods adopt a *Syn*PspA rod-like symmetry and orientation and cover a large range of diameters[2]. It has been suggested that columns of helix α0 in the lumen of Vipp1 rings stabilize the ring structure versus rod formation[1]. Nevertheless, the same or similar intermolecular interactions of α0 with another α0 and α1 + α4, as observed previously

in Vipp1 rings[1,3], are present in type I and II tubes. In the PspA-like Vipp1 (α1–α6) rods, however, a register shift and a larger distance to the next layer of monomers would make it impossible for α0 to reach α1 + α4 of the monomer in the adjacent layer. Thus, the presence and absence of α0 coordinates different assembly states of Vipp1 and interactions of helix α0 stabilize particular polymeric Vipp1 structures. Our 3.0-Å Vipp1 tube structure now reveals that D5, R6, R9, R12 and E23 of α0 interact with three different subunits through α4 (R170), α4 (K174) and α1 (D35) to hold these helices in place, thereby aligning the interacting subunits and helices in an orientation favoring rings and type I and II tubes over the PspA-like assembly, in addition to described lipid interaction properties.

In addition to the well-ordered symmetrical structures, we were able to determine the structure of membrane-bound Vipp1 carpets on the molecular level. The determined low-resolution Vipp1 carpet structures were reminiscent of wider Vipp1 rings, presumably with an approximately 90° rotation of the hairpin relative to the membrane axis. Formation of Vipp1 carpets is supposed to have an important role in Vipp1 membrane stabilization and protection and has been observed on flat solid-supported bilayers in vitro and presumably on TMs in vivo[12,14,24]. Additionally, in vivo observations of Vipp1 rods or tubes and (presumably) carpets but not rings suggest that these are equally physiologically relevant or stable forms of Vipp1 polymers[1,12,14]. We identified Vipp1 carpet assemblies on low-curvature and high-curvature membranes, while we found higher Vipp1 coverage on high-curvature membranes, in line with the observation that Vipp1 prefers binding to highly curved membranes (Fig. 6). Strikingly, our subtomogram averages of the carpet structures suggest a monomer arrangement similar to that found in Vipp1 rings, except that the assemblies do not close into a ring. Instead, the lattice is relaxed to adapt to lower membrane curvatures and extended to cover larger areas. The abundance of membrane-bound Vipp1 structures (loose coats, patches, carpets and rings) increased with increasing membrane curvature and reached a plateau at a curvature corresponding to an approximate vesicle diameter of 30 nm. Higher-diameter (lower-curvature) membranes are less frequently occupied with Vipp1 assemblies, in line with the observation that membrane-bound Vipp1 is concentrated at high-curvature regions of the TM in vivo[12]. Together, we observed that the local Vipp1 molecule concentration on the membrane affects the assembly state and thereby critically modulates membrane curvature (and vice versa).

In light of our findings, we propose the following sequence for Vipp1 membrane interactions (Fig. 7b): (i) Vipp1 monomers or small oligomers bind to and (ii) accumulate on low-curvature membranes in loose coats until a critical concentration is reached locally. (iii) At these membrane areas, Vipp1 monomers oligomerize in the plane of the membrane and the formed assembly patches start to induce curvature on the membrane, ultimately resembling the determined carpet structures. As Vipp1 preferentially binds to curved membranes (that is, membrane binding induces curvature), (iv, v) Vipp1 bulges start to grow on the membrane increasing the local curvature, stacking oligomers away from the membrane and thereby internalizing the membrane through helix α0 interactions. (vi) In this way, the initial bulges further emerge into the Vipp1 ring and (vii–x) stacked rings and subsequently elongate into rods that finally bud off the membrane. The transition of the emerging Vipp1 assembly is accompanied by a changed orientation of the Vipp1 hairpin α1–α3 from 90° at the low-curvature carpet to approximately 0° at the highest curvature with respect to the emerging tube axis. The hourglass shape of the stacked rings promote the transition from a flat membrane geometry to a highly curved membrane. When the Vipp1 assemblies constrict the internalized membrane beyond a certain threshold, the host membrane is pinched off and the assemblies can leave the membrane surface. Additionally, rings and short rods that budded off the membrane still contain highly curved membranes at their ends, prone to fuse with available free membranes to spawn new Vipp1 assembly sites or with other rings to form stacked rings. The driving forces for this process are determined by the tendency of Vipp1 to form highly ordered tubular lattices upon membrane interaction, thereby stabilizing curved membranes. As our data lack any temporal resolution, we cannot exclude that the process runs in the other direction or back and forth in both directions depending on the exact experimental conditions. In fact, previous studies suggested that Vipp1 rings in solution bind to membranes, subsequently disassemble[8] and transform into carpet structures[24]. Time-resolved structural research will be required to delineate the temporal order and other short-lived intermediates to comprehensively describe the structural states involved in this Vipp1-mediated membrane remodeling process.

## Online content

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

## Methods

### Expression and purification of Vipp1

Full-length Vipp1 (*sll0617*) of *Synechocystis* sp. PCC 6803 and associated mutants (Vipp1 (α1–α6) (truncation of helix α0 (aa 1–23)), Vipp1 dL10 (truncation of aa 157–166) and Vipp1 dL10Ala (substitution of aa 157–166 to alanine)) were heterologously expressed in *E. coli* C41 cells in terrific broth medium using a pET50(b) derived plasmid with a C-terminal His-tag and 3C protease cleavage site. For purification of Vipp1 and Vipp1 (α1–α6) under denaturing conditions, cells were resuspended after protein expression in lysis buffer containing 6 M urea (10 mM Tris-HCl pH 8.0 and 300 mM NaCl) supplemented with a protease inhibitor. Cells were lysed in a cell disruptor (TS Constant Cell disruption systems 1.1 kW; Constant Systems). The crude lysate was supplemented with 0.1% (v/v) Triton X-100 and incubated for 30 min at room temperature. Subsequently, the lysate was cleared by centrifugation for 15 min at 40,000*g*. The supernatant was applied to Ni-NTA agarose beads. The Ni-NTA matrix was washed with lysis buffer and lysis buffer with additional 10 mM imidazole. The protein was eluted from the Ni-NTA with elution buffer (10 mM Tris-HCl pH 8.0, 1,000 mM imidazole and 6 M urea). The fractions containing protein were pooled, concentrated (Amicon Ultra-15 centrifugal filter, 10-kDa molecular weight cutoff (MWCO)) and stored at −20 °C. For cleavage of the C-terminal His-tag, the protein was diluted 1:1 in cleavage buffer (10 mM Tris-HCl, 300 mM NaCl, 1 mM DTT and 0.1 mM EDTA) and dialyzed overnight against cleavage buffer together with 3C protease, including three buffer exchanges. After the cleavage reaction, the protein was diluted 1:3 with 10 mM Tris-HCl pH 8.0, 8 M urea, 200 mM NaCl and 7 mM imidazole and applied to the Ni-NTA matrix. The Ni-NTA matrix was washed with 10 mM Tris-HCl pH 8.0, 6 M urea, 100 mM NaCl and 4 mM imidazole. Flowthrough and wash fractions containing the cleaved protein were pooled, concentrated (Amicon Ultra-15 centrifugal filter, 10-kDa MWCO) and stored at −20 °C. Protein concentrations were determined by measuring the absorbance at 280 nm of Vipp1 diluted in 4 M guanidine hydrochloride using the respective molar extinction coefficient calculated by ProtParam[35].

### Liposome preparation and lipid reconstitution

Chloroform-dissolved EPL extract was purchased from Avanti Polar Lipids. Lipid films were produced by evaporating the solvent under a gentle stream of nitrogen and vacuum desiccation overnight. For Vipp1 reconstitution with lipids, urea-unfolded Vipp1 was added to the EPL film (Extended Data Table 1) and incubated with shaking for 20–30 min at 37 °C until the lipid film was resolved. Subsequently, the mixture was dialyzed overnight against 10 mM Tris-HCl pH 8.0 (8 °C, 20-kDa MWCO) including three buffer exchanges to refold the protein.

For analyzing membrane binding of Vipp1 helix α0 peptides, buffer was added to the EPL film to a final lipid concentration of 8 mM and incubated at 37 °C with shaking for 30 min. The liposomes were then prepared by sonication (Branson tip sonifier).

**Characterization of the refolded protein.** Vipp1 prepared as described above was refolded by dialysis against 10 mM Tris-HCl pH 8.0 overnight, with three buffer exchanges. Protein concentrations were determined on the basis of protein absorption at 280 nm as described above. For comparison, Vipp1 purified under native conditions (as described previously[13]) was studied. The protein concentration was adjusted to match the concentrations of the refolded protein on the basis of a comparison of the tryptophan fluorescence intensity at 340 nm determined in 6 M urea. The thermal stability was studied by monitoring the change in secondary structure through CD spectroscopy (J-1500, Jasco). The CD spectrum between 210 and 230 nm was measured at different temperatures (2 °C temperature steps, 1-nm wavelength steps, overall mean temperature ramp of 10 °C h$^{-1}$). Samples of each protein were prepared in duplicate and measured in parallel in a six-cell holder. Interaction of the refolded protein with preformed liposomes was monitored by following changes in tryptophan fluorescence. To this end, an EPL film, prepared as described above, was rehydrated with Tris buffer. Unilamellar liposomes were either formed by sonication (Branson tip sonifier) to yield small liposomes or by extrusion through a 200-nm filter (Avanti Polar Lipids). The size of the liposomes was determined by dynamic light scattering (Zetasizer Pro, Malvern). According to the *z* average, the extruded liposomes had a diameter of 143 ± 0.5 nm and the sonified liposomes had a diameter of 56 ± 0.44 nm. A concentration of 3 μM protein was incubated with liposomes for 45 min at room temperature. Fluorescence emission was determined from 300 nm to 450 nm at 25 °C after excitation at 280 nm, with both excitation and emission slit widths set to 2.5 nm (FP-8500, Jasco).

**Membrane binding of Vipp1 helix α0 peptides.** The peptides Vipp1 helix α0 WT (MGLFDRLGRVVRANLNDLVSKAED), Vipp1 helix α0 R6A;R9A (MGLFDALGAVVRANLNDLVSKAED), Vipp1 helix α0 F4A;V11A (MGLADRLGRVARANLNDLVSKAED) and Vipp1 helix α0 amph(−) (MGAADRAGRAARANANDAASKAED) were purchased from PSL. The changes in secondary structure upon membrane binding were determined by CD spectroscopy (J-1500, Jasco). Then, 40 μM peptide (10 mM Tris pH 8.0) was incubated with increasing amounts of liposomes (up to 4 mM EPL). After incubation at room temperature for 15 min, spectra were recorded between 190 and 250 nm in 1-nm steps at 25 °C. Three individual samples were measured for each sample, with 12 accumulations per sample. The mean and s.d. of the value from the three samples at each lipid concentration were used to create a binding curve.

### Negative-staining EM

For negative-staining EM, 3 μl of the sample was applied to glow-discharged (PELCO easiGlow glow discharger, Ted Pella) continuous carbon grids (CF-300 Cu, EM Sciences). The sample was incubated on the grid for 1 min. Then, the grid was side-blotted using filter paper, washed with 3 μl of water, stained with 3 μl of 2% uranyl acetate for 30 s and air-dried. The grids were imaged with a 120-kV Talos L120C EM instrument (Thermo Fisher Scientific, FEI) equipped with a CETA camera at a pixel size of 2.49 Å per pixel (×57,000 magnification) at a nominal defocus of 1.0–2.5 μm.

### Cryo-EM

Grids were prepared by applying 4 μl of sample (with or without 5-nm gold fiducials) (Extended Data Table 1) to glow-discharged (PELCO easiGlow glow discharger, Ted Pella) Quantifoil grids (R1.2/1.3 Cu 200 mesh, EM Sciences). The grids were plunge-frozen in liquid ethane using a Leica EM GP2 plunge freezer set to 80% humidity at 10 °C (sensor-guided backside blotting, blotting time of 3–5 s). Videos were recorded in underfocus on a 200-kV Talos Arctica G2 (Thermo Fisher Scientific) EM instrument equipped with a Bioquantum K3 (Gatan) detector or a 300 kV Titan Krios G4 (Thermo Fisher Scientific) EM instrument equipped with a Biocontinuum K3 (Gatan) and a Falcon4i (Thermo Fisher Scientific) detector operated by EPU (Thermo Fisher Scientific) or Tomo (Thermo Fisher Scientific). Tilt series for cryo-electron tomography (cryo-ET) were collected at −60° to 60° with 3° in a dose-symmetric scheme. Tilt images were acquired at a magnification of ×53,000 (pixel size of 1.7 Å) with a nominal underfocus of 3.0 μm. The total dose for each tilt series was 152 e$^-$ per Å$^2$. A total of 176 tilt series were collected (details of the single-particle and cryo-ET acquisitions in Tables 1 and 2 and Extended Data Table 1).

### Single-particle image processing and helical reconstruction

Video frames were gain-corrected, dose-weighted and aligned using cryoSPARC Live[36]. Initial 2D classes were produced using the autopicker implemented in cryoSPARC Live. The subsequent image-processing steps were performed using cryoSPARC. The classes with most visible detail were used as templates for the filament tracer. For the Vipp1 EPL dataset, the resulting filament segments were extracted with a

600-pixel box size (Fourier-cropped to 200 pixels) and subjected to multiple rounds of 2D classification. The remaining segments were sorted by filament class: (1) stacked rings; (2) type I tubes; and (3) type II tubes. For each filament class, the remaining segments were re-extracted with a box size of 400 pixels (Fourier-cropped to 200 pixels) and subjected to an additional round of 2D classification. The resulting 2D class averages were used to determine filament diameters and initial symmetry guesses in PyHI[37]. Initial symmetry estimates were validated by helical refinement in cryoSPARC and selection of the helical symmetry parameters yielding reconstructions with typical Vipp1 features and the best resolution. Subsequently, all segments for each filament class were classified by heterogeneous refinement and followed by 3D classifications using the initial helical reconstructions as templates. The resulting helical reconstructions were subjected to multiple rounds of helical refinement including the symmetry search option. Reconstructions of the stacked rings were treated similarly; however, instead of helical symmetry, the respective rotational symmetry was applied in helical reconstruction jobs (C11–C14). Segments with poor visible details were discarded by heterogeneous refinement. For the final polishing, the segments were re-extracted at 600 pixels with Fourier cropping to 400 pixels (stacked rings and type I tubes) or at 500 pixels without Fourier cropping (type II tubes). The Vipp1 (α1–α6) and Vipp1 (α1–α6) EPL datasets were preprocessed as described above. The filament segments were extracted with a 700-pixel box size (Fourier-cropped to 200 pixels) and subjected to multiple rounds of 2D classification. The subsequent image-processing steps were identical to the above-described workflow of helical reconstruction of type I and type II filaments. Bad segments were discarded by heterogeneous refinement. For the final polishing, the segments were re-extracted at 700 pixels with Fourier cropping to 350 pixels.

The Vipp1 dL10Ala EPL dataset was preprocessed using WARP[38] (motion correction, binning to physical pixel size and contrast transfer function (CTF) estimation) and further processed in cryoSPARC as described above. The filament segments were extracted with a 500-pixel box size (Fourier-cropped to 200 pixels) and subjected to multiple rounds of 2D classification. The subsequent image-processing steps were identical to the above-described workflow of helical reconstruction of type I and type II filaments. Poor segments were discarded by heterogeneous refinement or 3D classification. For the final polishing, the segments were re-extracted at 500 pixels without Fourier cropping. The ring structures were picked using a template picker and extracted at 600 pixels with Fourier cropping to 200 pixels. The extracted particle stack was cleaned after multiple rounds of 2D classification. Rings with different rotational symmetries were sorted by multiclass ab initio reconstruction and further classified by multiple rounds of heterogenous refinement and ab initio reconstruction with imposed rotational symmetry (C15–C20). Poor particles were discarded by heterogeneous refinement. For the final polishing, the particles were re-extracted at 600 pixels with Fourier cropping to 450 pixels and subjected to nonuniform refinement with imposed symmetry (C15–C20). The local resolution distribution and local filtering for all maps were performed using cryoSPARC (Extended Data Figs. 2a, 3a, 4e, 6d and 7a). The resolution of the final reconstructions was determined by Fourier shell correlation (FSC; automasked, 0.143).

## Cryo-ET image processing
Tilt image frames were gain-corrected, dose-weighted and aligned using WARP[38]. The resulting tilt series were aligned, 8× binned and reconstructed by the weighted backprojection method using AreTomo[39]. For segmentation, the reconstructed tomograms were filtered with a recursive exponential filter followed by a nonlocal means filter using Amira (Thermo Fisher Scientific). The filtered tomograms were segmented in Dragonfly (Object Research Systems) by progressively training a U-Net with an increasing number of manually segmented tomogram frames (5–15). Then, the trained U-Net was used

to predict those features of the tomogram. For visualization, the segmentation was cleaned up by removing isolated voxels of each label group (islands of <150 unconnected voxels). The resulting segmentation was imported to ChimeraX[40] for 3D rendering. For quantitative analysis of membrane curvature, eight tomograms were used in the same manner to train the U-Net, which was then used to predict 123 tomograms included in the analysis. Once the resulting segmentations were coarsely corrected for errors, they were exported as .tif files and assembled into .mrc files for further analysis. For the membrane segmentations, the surface morphometrics toolkit[41] was used to determine curvature and membrane normals. The resulting coordinates and their corresponding curvature estimates were used to filter for adjacent carpet segmentations, accepting a distance up to 50 Å. To minimize errors, coordinates close to tomogram edges were excluded. For further analysis, coordinates were sorted into fixed-size bins according to their curvedness and, for each bin, the ratio of membrane coordinates with a present proximal carpet segmentation was determined resulting in the corresponding occupancy value between 0 and 1 (0, no coverage; 1, full coverage).

For subtomogram averaging, RELION 4 (ref. [42]) was used. CTF estimation was conducted with ctffind4 (ref. [43]). As initial particles, the coordinates resulting from the previously described membrane analysis were used. Coordinates were selected for the presence of Vipp1 carpet structures (within a radius of 100 Å) and a minimal interparticle distance of 15 Å was enforced to avoid excessive overlaps. The orientation of particles was initially estimated using their corresponding membrane normal to restrain angular searches for all subsequent refinement procedures. To create an initial model from a subset of homogenous particle picks, all coordinates were clustered according to their corresponding membrane curvature, using the Scikit-Learn implementation of k-means. After particle averaging and several rounds of 3D classifications and refinements in 4× binning, a low-resolution model emerged. This model was then used as an initial reference and, from here on, all particles were included. Multiple rounds of 3D classifications and refinements for cleaning up the dataset followed, initially using 4× binning and later on 2× binning. As a further means to exclude poor particles, particles with diverging orientations compared to the majority of particles in close proximity were excluded. Finally, the resulting maps were used to improve predetermined parameters, using RELION's frame alignment and CTF refinement procedures followed by a final round of refinements. For the highest-resolution map, the average curvedness of all points from which the final average was calculated (-0.054) indicated a substantially higher diameter than the currently existing highest diameter Vipp1 ring model (C18, Protein Data Bank (PDB) 7O3Z). Therefore, PDB 7O3Z was used to initially set the distance to the symmetry axis and rotation. To find the correct symmetry parameters (C symmetry, distance and rotation in relation to the symmetry axis), the map was fitted in ChimeraX[40] along the x axis. The correlation of the original map with versions of itself with different symmetry parameters applied was then calculated and the correlation was maximized. First, translational searches along the x axis in 1-Å increments were conducted for C18–C22 symmetries. The best correlation for the 0.054 curvedness reconstruction was identified for a C20 symmetry. When optimal distance and symmetry parameters were found, rotational searches were conducted (Supplementary Fig. 6e). Finally, the symmetry, shift and rotational parameters were applied to the reconstruction and PDB 7O3Z was rigid-body fitted into the resulting map.

## Cryo-EM map interpretation and model building
The handedness of the maps was determined by rigid-body fitting Vipp1 reference structures (PDB 7O3W) into the final maps using ChimeraX[40,44] and flipped accordingly. For models of the Vipp1 type I and type II and Vipp1 dL10Ala type IIb tubes, one of the monomers of PDB 7O3W underwent molecular dynamics flexible fitting (MDFF) to

the 3D reconstructions using ISOLDE[45]. The models were subjected to autorefinement with 'phenix.real_space_refine' (ref. 46). The autorefined models were checked and adjusted manually in Coot[47] and ISOLDE[45] before a final cycle of autorefinement with 'phenix.real_space_refine' (ref. 46). After the final inspection, the models were validated using 'phenix.validation_cryoem' (ref. 48) and MolProbity[49]. Then, the respective helical symmetry was applied to all models to create assemblies of 60 monomers each to generate 'biomt' matrices for deposition. For models of the stacked *C11* and *C12* rings PDB 6ZVR and PDB 6ZVS were used as ref. 3. Side chains were adjusted to the sequence of *Synechocystis* sp. PCC 6803 Vipp1 and the resulting models were rigid-body fitted to the maps. For models of the stacked *C13* and *C14* rings, PDB 7O3W was used as a ref. 1. The nucleotide was deleted and the monomers were first rigid-body fitted and then underwent MDFF to the 3D reconstructions using ISOLDE[45]. For models of the Vipp1 (α1–α6) rods, PDB 7O3W was used as a ref. 1. Only the central monomer was selected and aa 1–22 were removed. The truncated model was rigid-body fitted to the maps. For each map, the monomers then underwent MDFF using ISOLDE[45]. The models were subjected to autorefinement with 'phenix.real_space_refine' (ref. 46). The autorefined models were checked and adjusted manually in Coot[47] and ISOLDE[45] and subjected to a final cycle of autorefinement with 'phenix.real_space_refine' (ref. 46). After the final inspection, the models were validated using 'phenix.validation_cryoem' (ref. 48) and MolProbity[49]. Then, the respective helical symmetry was applied to all models to create assemblies of 60 monomers each to generate 'biomt' matrices for deposition. The Vipp1 dL10Ala *C15* to *C20* ring complex models were generated using Vipp1 WT *C15* and *C16* ring models as references (PDB 7O3X and 7O3Y)[1]. The nucleotide was removed and the monomers were first rigid-body fitted and then underwent MDFF to the 3D reconstructions using ISOLDE[45]. Image processing, helical reconstruction and model building were completed using SBGrid-supported applications[50]. In this manner, from a total of 37 cryo-EM maps determined in four samples (3× Vipp1 + EPL rods, 4× Vipp1 + EPL stacked rings, 10× Vipp1 (α1–α6) rods, 10× Vipp1 (α1–α6) + EPL rods, 4× Vipp1 dL10Ala EPL tubes and 6× Vipp1 dL10Ala rings), a total of 17 unique PDB coordinates were refined and deposited originating from four samples (Table 1). The four Vipp1 + EPL stacked-ring maps and the six Vipp1 dL10Ala ring complex maps were not of sufficient quality to refine models. The remaining complementary Vipp1 (α1–α6) + EPL maps were of identical symmetry consisting of either similar or poorer resolution densities; therefore, atomic model refinement was not pursued further. Diameters of the Vipp1 rod and tube reconstructions were measured as radial density profiles using ImageJ[51]. Outer and inner leaflet radii of engulfed membrane tubes were determined from the same radial density profiles at the peak maxima from each bilayer leaflet.

## Quantification and statistical analysis

Data and statistical analysis were performed using OriginPro 2022b (OriginLab). Detailed descriptions of quantifications and statistical analyses (exact values of *n*, dispersion and precision measures used and statistical tests used) can be found in the respective figures and figure legends and the Methods.

## Reporting summary

Further information on research design is available in the Nature Portfolio Reporting Summary linked to this article.

## Data availability

All unique and stable reagents generated in this study are available from the lead contact with a completed material transfer agreement. The cryo-EM maps are available from the EMDB under accession numbers EMD-18384, EMD-18421, EMD-18420, EMD-18422, EMD-18423, EMD-18424, EMD-18425, EMD-18426, EMD-18427, EMD-18428, EMD-18429, EMD-18430, EMD-18431, EMD-18432, EMD-18433, EMD-18434, EMD-18435, EMD-18620, EMD-19863, EMD-19864, EMD-19865, EMD-19866, EMD-19899, EMD-19900, EMD-19901, EMD-19902, EMD-19903 and EMD-19904. The Vipp1 models are available from the PDB under accession numbers 8QFV, 8QHW, 8QHV, 8QHX, 8QHY, 8QHZ, 8QI0, 8QI1, 8QI2, 8QI3, 8QI4, 8QI5, 8QI6, 9EOM, 9EON, 9EOO and 9EOP. Datasets accessed in this manuscript are available from the PDB under accession numbers 7O3Y, 7O3X, 7O3W and 7O3Z. Source data are provided with this paper.

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

## Acknowledgements

This study was funded by the Deutsche Forschungsgemeinschaft (SA 1882/6-1, SCHN 690/16-1, CRC1208 (project no. 267205415, to C.S.) and CRC1551 (project no. 464588647, to D.S.)). D.K. and

C.S. acknowledge support of the project KA1-Co-02 'CoViPa', a grant from the Helmholtz Association's Initiative and Network Fund. We gratefully acknowledge the EM training, imaging and access time granted by the life science EM facility of the Ernst-Ruska Center at Forschungszentrum Jülich. The authors gratefully acknowledge the computing time granted by the JARA Vergabegremium and provided on the JARA Partition part of the supercomputer JURECA at Forschungszentrum Jülich[52].

## Author contributions

B.J., D.S. and C.S. designed the research. I.R. cloned, expressed and purified the proteins. B.J. prepared cryo-EM samples and operated the EM instruments. B.J. and C.S. determined the cryo-EM structures. B.J. built the refined atomic models. B.J. and D.K. reconstructed and segmented the tomograms. D.K. determined the STA structures. B.J., D.K. and C.S. interpreted the protein structures. M.K. analyzed the peptides. N.H. analyzed the isolated proteins. M.K., N.H. and D.S. interpreted the membrane-binding data. B.J., D.K., D.S. and C.S. prepared the manuscript with input from all authors.

## Funding

## Competing interests

The authors declare no competing interests.

## Additional information

**Extended data** is available for this paper at https://doi.org/10.1038/s41594-024-01399-z.

**Correspondence and requests for materials** should be addressed to Carsten Sachse.

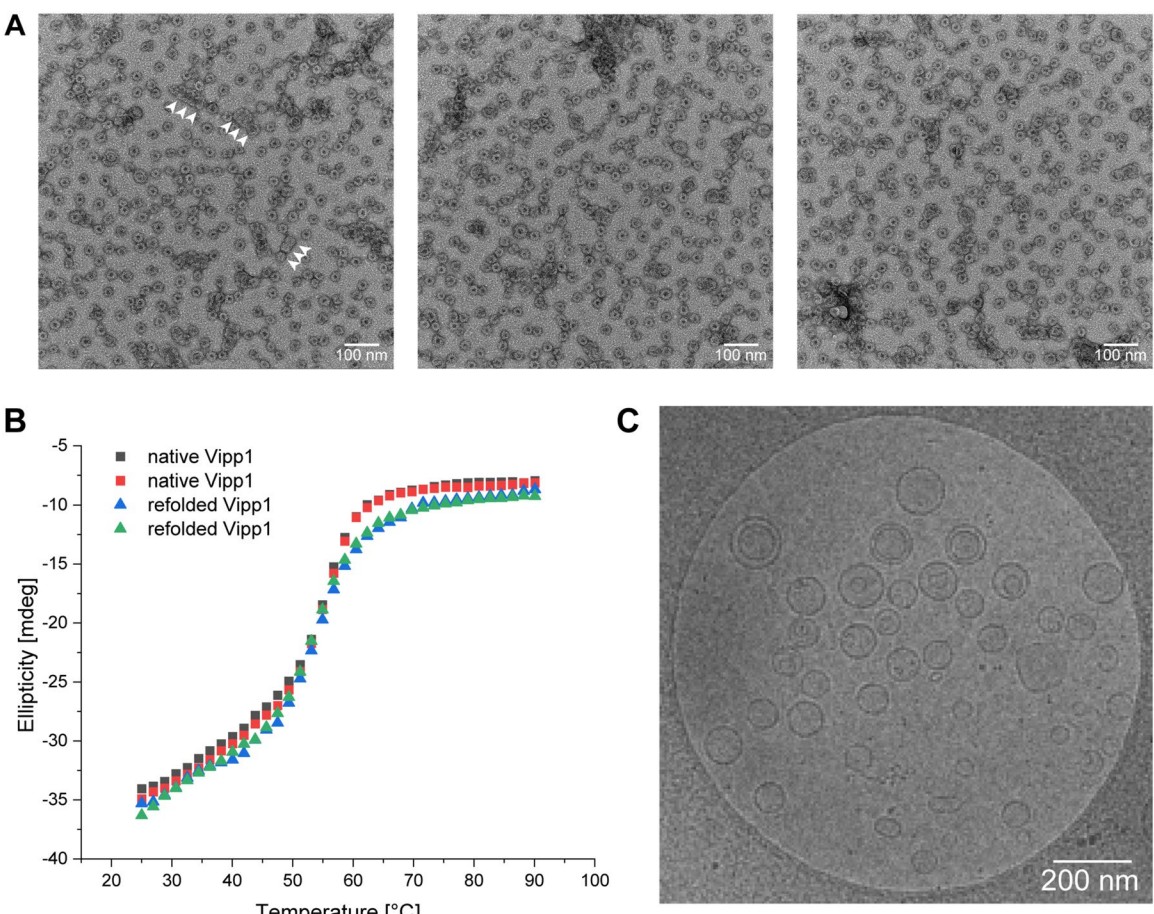

**Extended Data Fig. 1 | Negative staining EM of Vipp1 preparations. a**: Negative staining EM micrographs of Vipp1 in the absence of membranes (Dataset size 9 micrographs). In the absence of membranes, Vipp1 forms small oligomers, ring complexes, and stacked-ring assemblies (white arrowheads). **b**: Thermal stability of Vipp1 proteins. Vipp1 with His-tag purified under native conditions (black, red) or Vipp1 purified in the presence of 6 M urea (green, blue) was exposed to increasing temperatures. **c**: Cryo-EM micrograph of EPL liposomes without Vipp1 (Dataset size 126 micrographs).

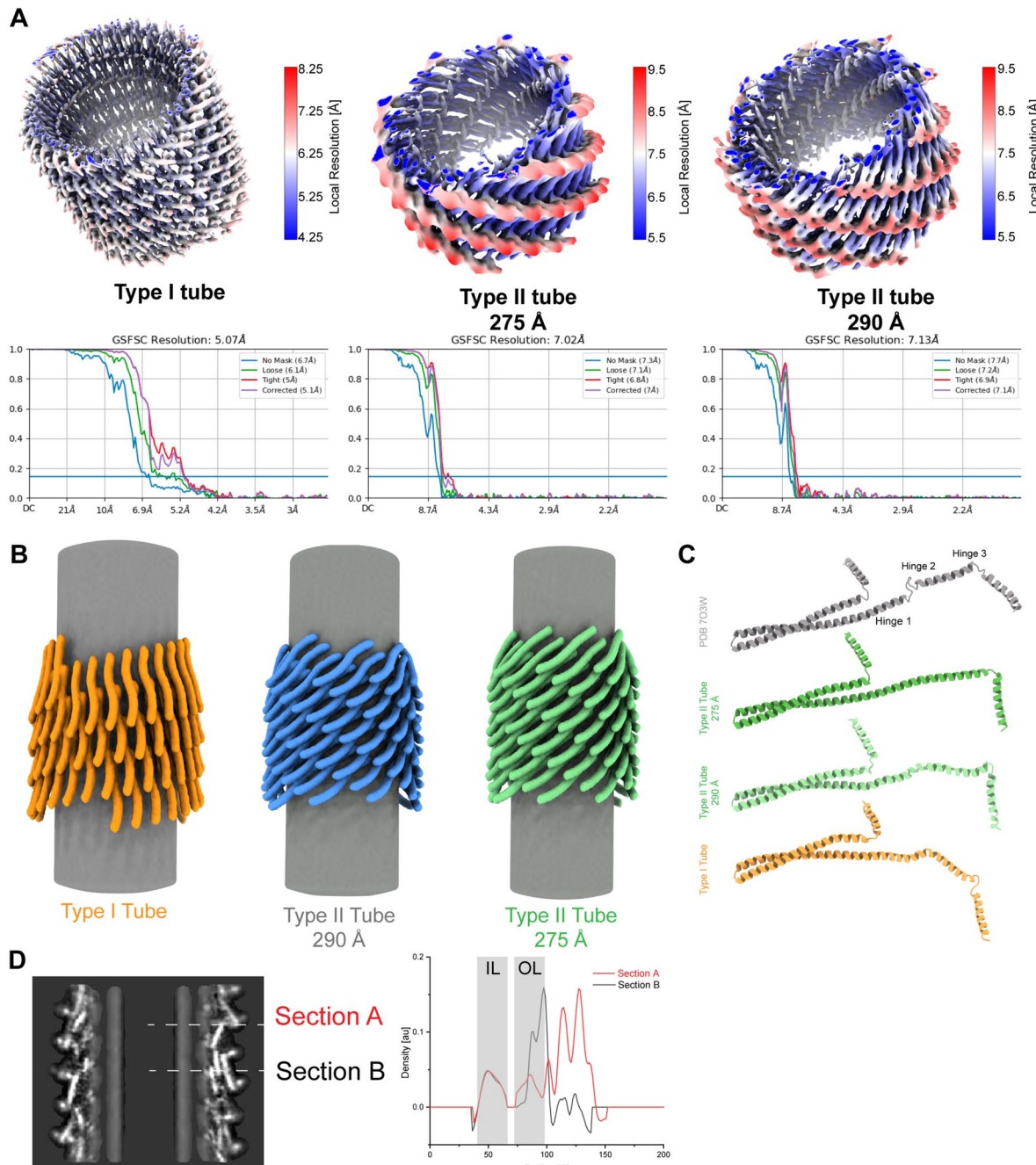

**Extended Data Fig. 2 | Local resolution maps and structural features of Vipp1 tubes. a**: Top: local resolution maps of Vipp1 tubes (FSC = 0.5). Color bars show the local resolution range in Å. Bottom: FSC curves with global resolutions. **b**: Simplified models of the Vipp1 tubes highlighting the orientation of the monomers with respect to the membrane axis. **c**: Comparison of the monomer structures extracted from the specified Vipp1 tubes. **d**: Xy-slice of the Type I tubes with sections at the outer leaflet far from helix α0 (Section A) and close to helix α0 (Section B) and the corresponding density profiles, showing the indentation in the outer leaflet caused by helix α0. IL: inner leaflet, OL: outer leaflet.

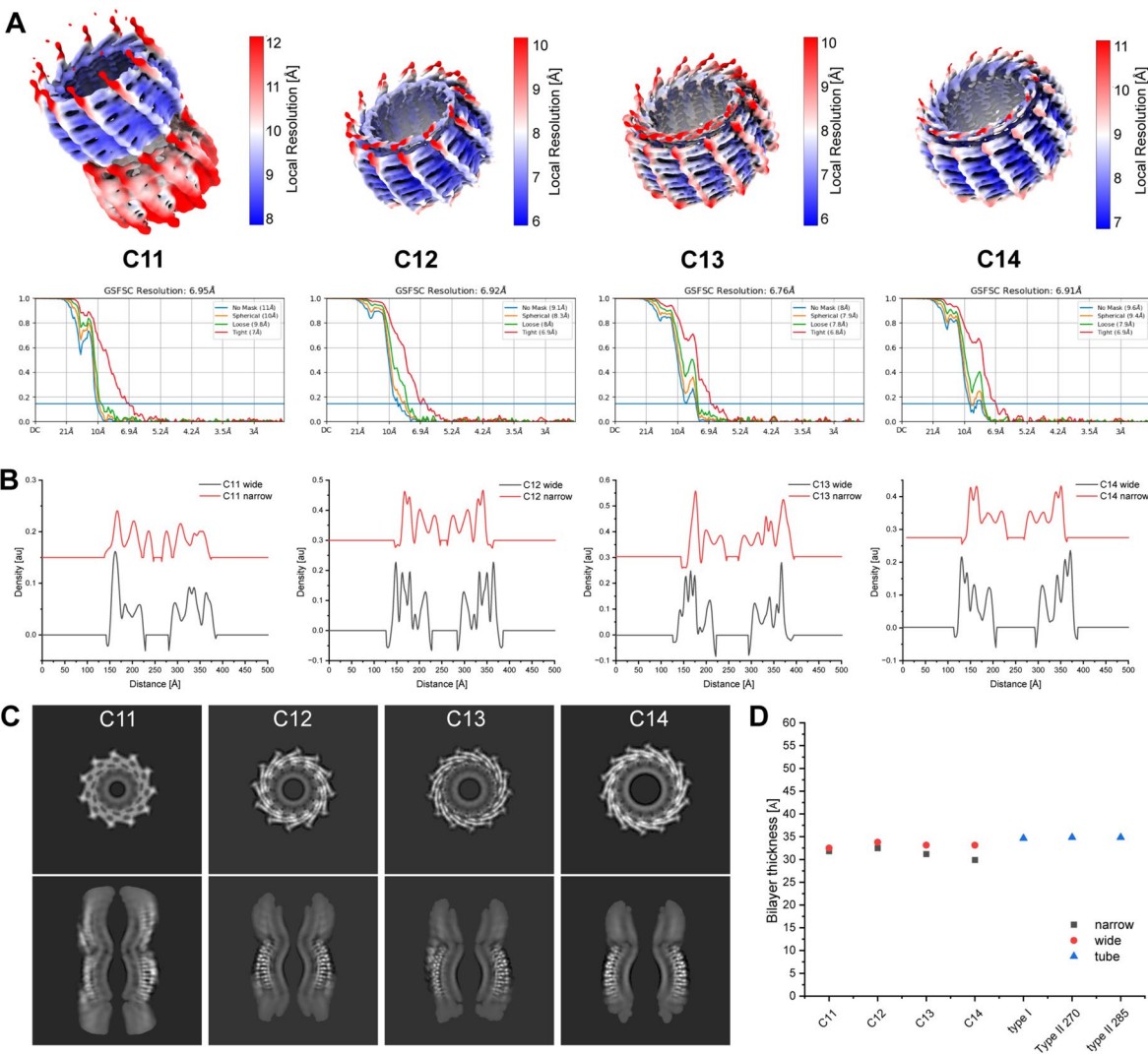

**Extended Data Fig. 3 | Local resolution and structural features of stacked Vipp1 rings with engulfed membrane. a**: Local resolution maps of Vipp1 stacked-ring assemblies (FSC = 0.5). Top: Color bars show the local resolution range in Å. Bottom: FSC curves with global resolutions. **b**: Density plots of Vipp1 rings with engulfed membrane. Red: Density profile at the narrowest part of the ring; black: density profile at the widest part of the ring. **c**: Z and xy-slices of stacked C11-C14 ring models in greyscale. **d**: Scatter plot of the bilayer thickness identified in different Vipp1 assemblies.

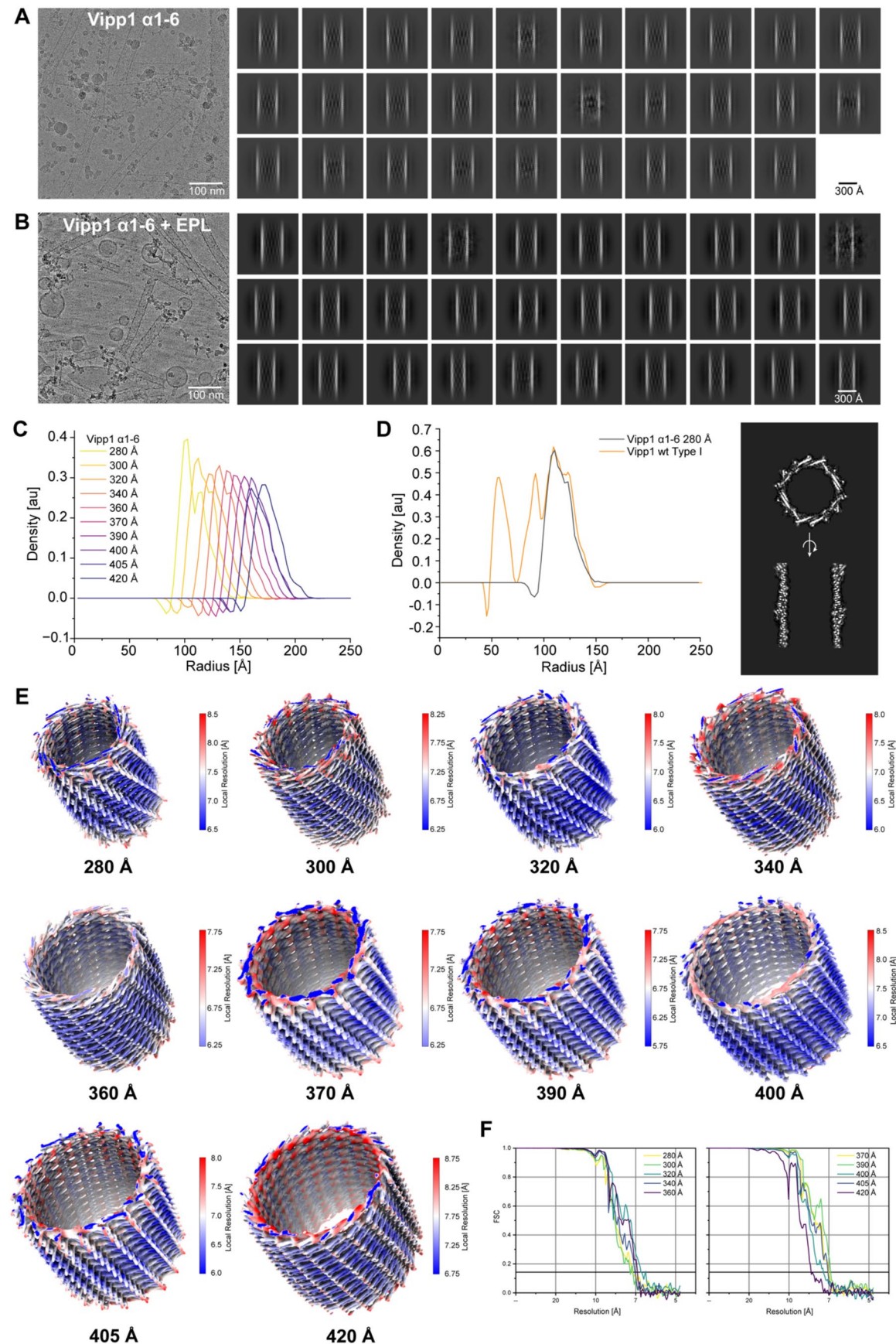

**Extended Data Fig. 4 | See next page for caption.**

**Extended Data Fig. 4 | Vipp1(α1-6) helical tubes in the absence and presence of membrane. a/b**: Cryo-EM micrographs and 2D class averages Vipp1(α1-6) with and without EPL membrane reconstitution (Dataset size 5,901 and 2,463 micrographs). **c**: Radial density plots of Vipp1(α1−6) rods with diameters from 280 to 420 Å. **d**: Left: radial density plots of Vipp1 Type I tubes and Vipp1(α1−6) 300 Å rods with lipids. Right: z (top) and xy-slice (bottom) of the Vipp1(α1−6) 300 Å rod map in greyscale. **e**: Local resolution maps of Vipp1(α1-6) rods (FSC = 0.5) with the corresponding diameters. Color bars show the local resolution range in Å. **f**: FSC curves of 10 Vipp1(α1-6) structures with global resolutions.

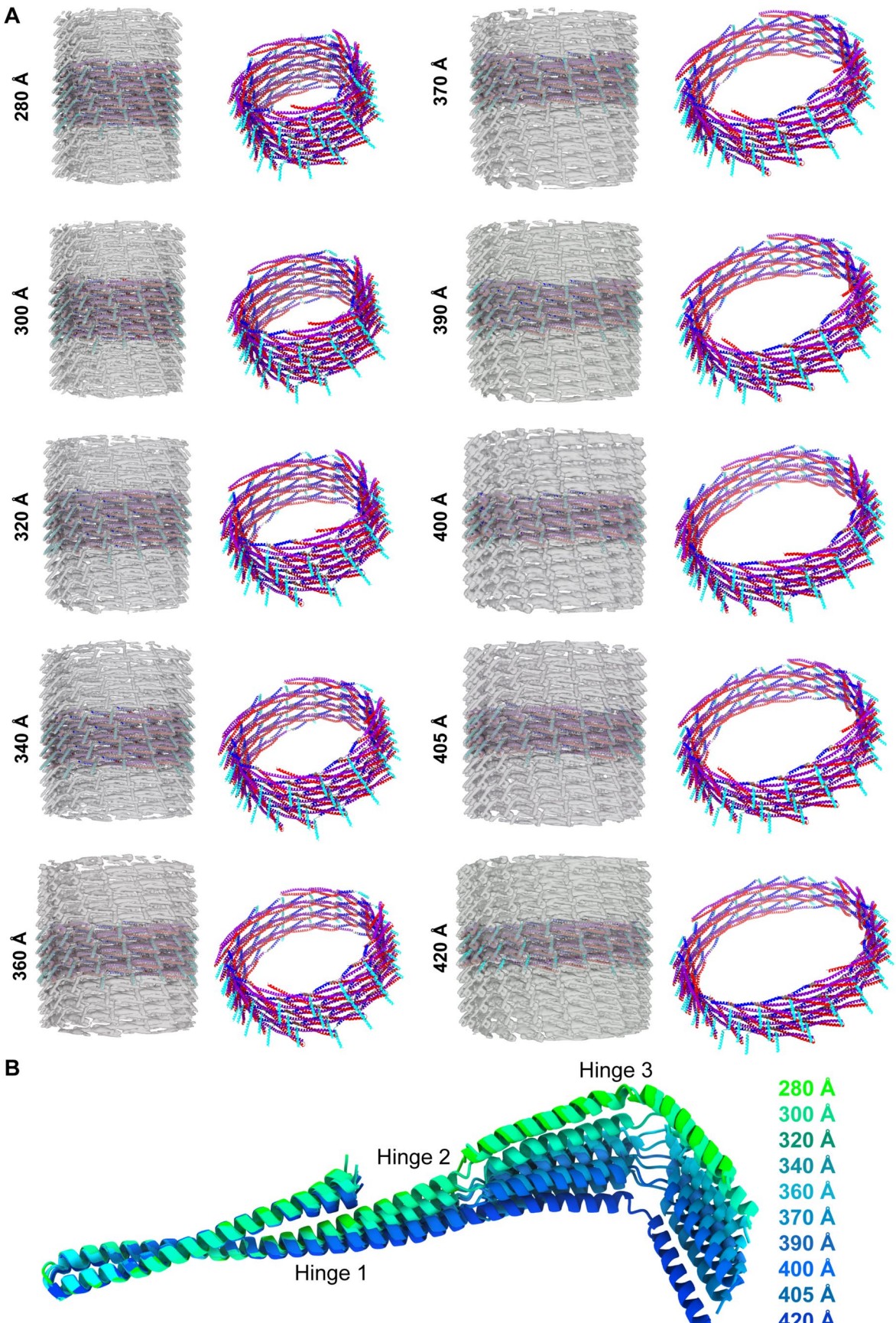

**Extended Data Fig. 5 | Vipp1(α1-6) helical tubes in the absence and presence of membrane. a**: Cryo-EM maps with fitted models (left) and corresponding atomic models (right) of Vipp1(α1–6) rods in ribbon representation (α1 red, α2 + 3 violet, α4 blue, α5 cyan). **b**: Superposition of the determined Vipp1(α1-6) conformations with the corresponding diameters.

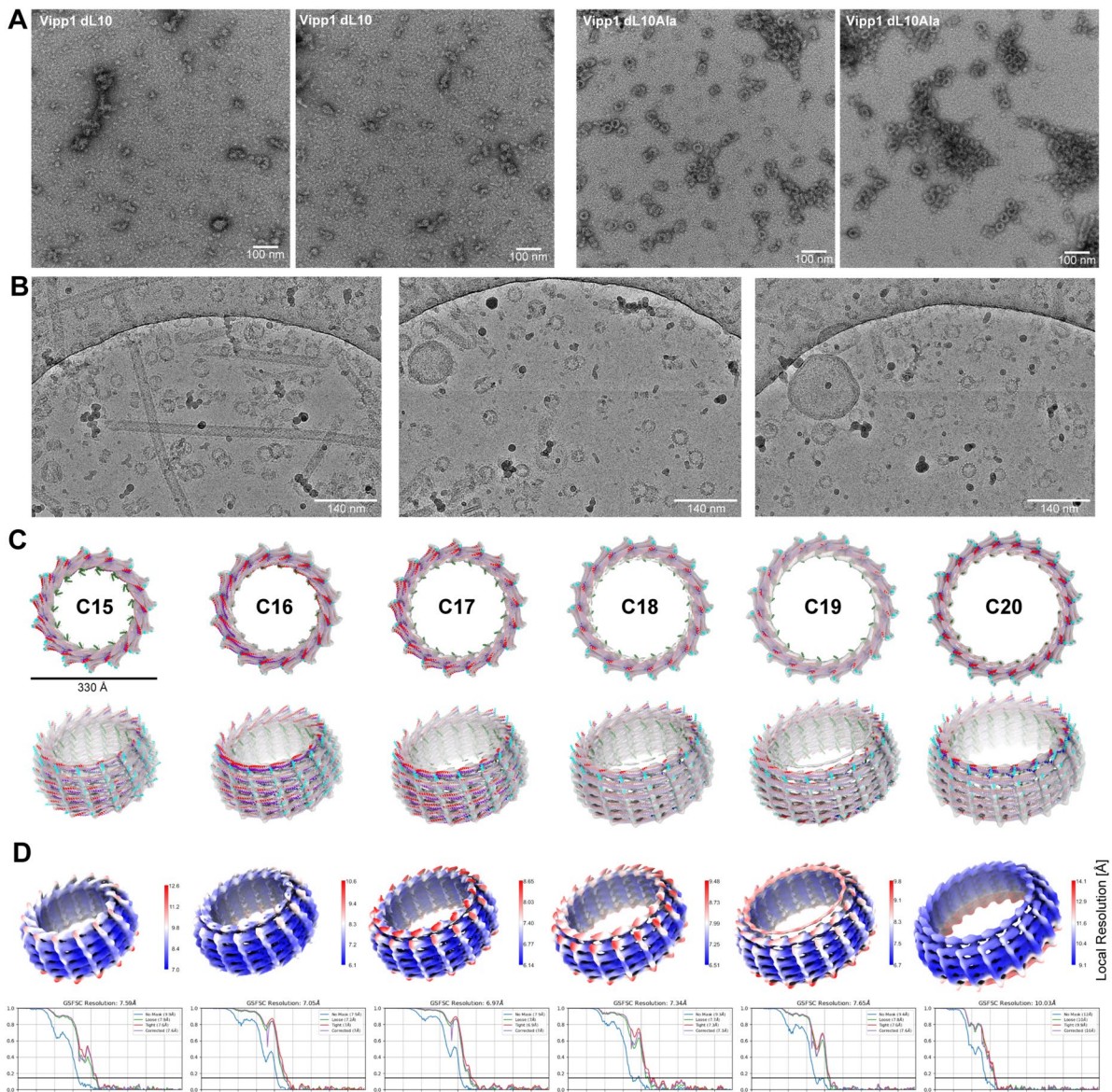

**Extended Data Fig. 6 | Vipp1(α1-6) helical tubes in the absence and presence of membrane. a**: Negative staining EM micrographs of Vipp1dL10 and Vipp1 dL10Ala in the absence of membranes (Dataset size 6 and 9 micrographs). Vipp1 dL10 forms small oligomers and larger irregular assemblies. Vipp1 dL10Ala forms small oligomers and ring complexes. **b**: Representative cryo-EM microgrpahs of the Vipp1 dL10Ala+EPL sample (Dataset size 29,548 micrographs). **c**: Cryo-EM maps and fitted models of C15 to C20 Vipp1 dL10Ala rings. **d**: Local resolution maps of Vipp1dlA10 rings and tubes (FSC = 0.5) with the corresponding diameters. Color bars indicate the mapped local resolution values in Å. FSC curves show the global resolution of the respective reconstruction at FSC = 0.143.

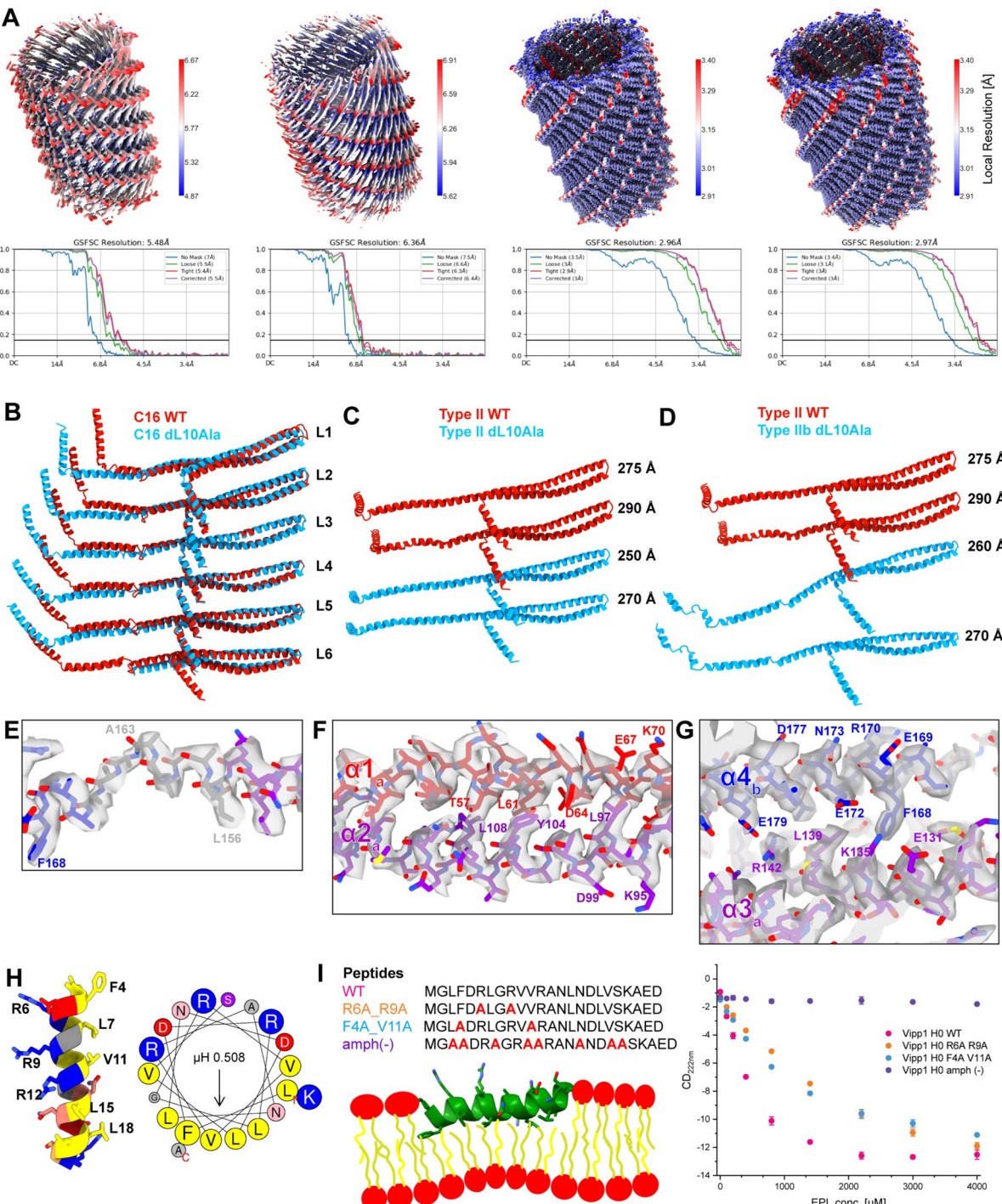

**Extended Data Fig. 7 | Vipp1(α1-6) helical tubes in the absence and presence of membrane. a**: Local resolution maps of Vipp1dlA10 tubes (FSC = 0.5) with the corresponding diameters. Color bars show the local resolution range in Å. FSC curves show the global resolution of the respective reconstruction at FSC = 0.143. **b**: Superposition of Vipp1 wild-type (WT) (PDB: 7O3Y) and Vipp1 dL10Ala monomers from C16 rings. **c**: Superposition of Vipp1 WT and Vipp1 dL10Ala monomers from Type II tubes. **d**: Superposition of Vipp1 WT and Vipp1 dL10Ala monomers from Type IIb tubes. **e-g**: Enlarged views of the cryo-EM map

with the fitted model of Vipp1 dL10Ala Type IIb tubes. **e**: Hinge 2, **f**: Intramolecular interactions within the hairpin, **g**: Intermolecular interactions of the $_{168}$FERM$_{171}$ motif. **h**: Helix-wheel plot of helix α0 (created with HeliQuest (Gautier et al., 2008)) **i**: Membrane binding of Vipp1 helix α0 peptides analyzed via CD spectroscopy with the indicated mutation and putative orientation of Vipp1 helix α0 in the membrane, as observed in Vipp1 tubes. (n = 3 (biological replicates), errors represent SD).

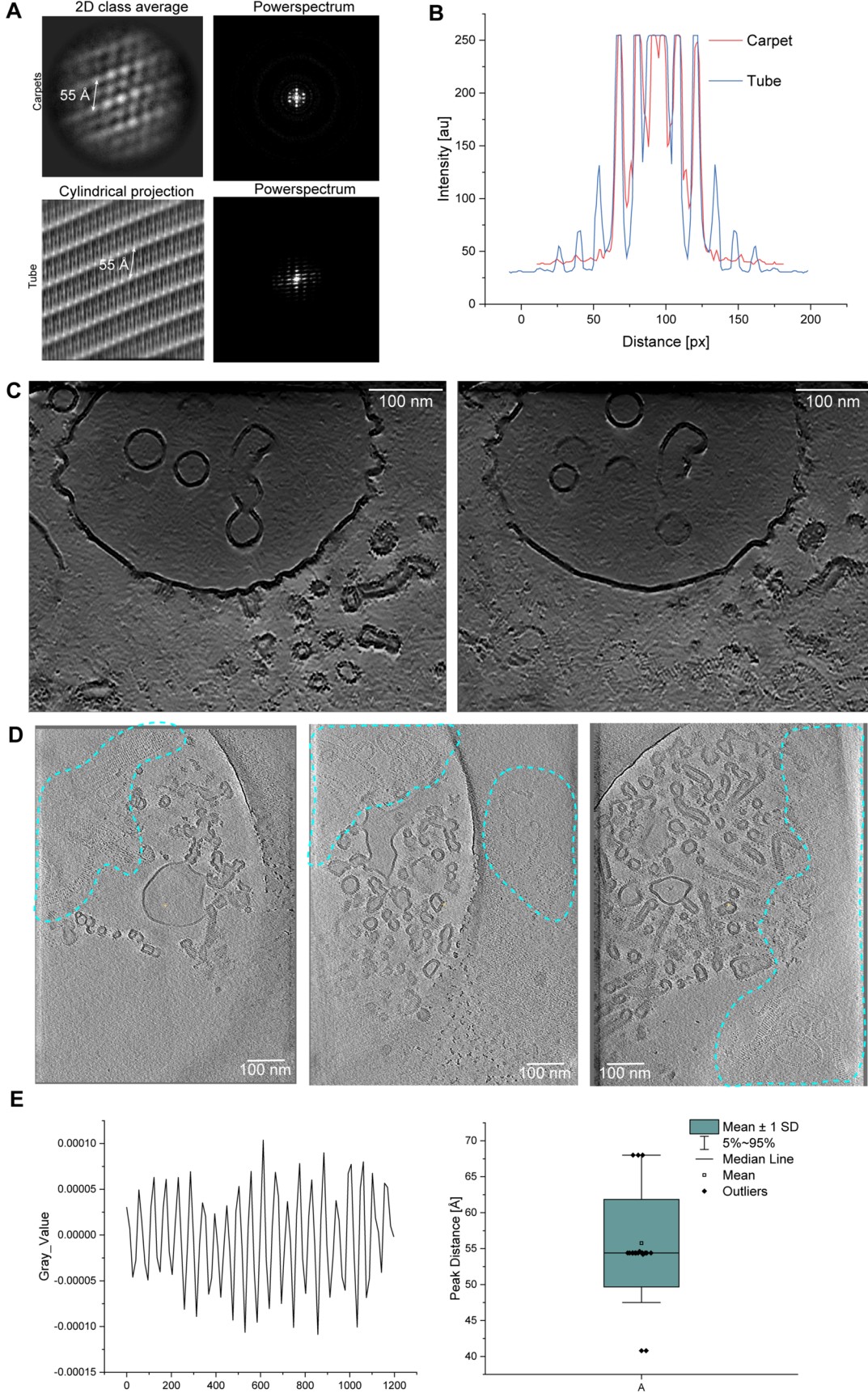

**Extended Data Fig. 8 | Vipp1 spirals and carpets. a**: Left: 2D class averages of the Vipp1 carpets and cylindrical projection of the Type I tubes. Right: corresponding power spectra of 2D class averages of the Vipp1 carpets and Type I tubes. **b**: Overlay of the meridional density plots of the power spectra from A.

**c**: Tomogram slices of Vipp1+EPL before segmentation. **d**: Tomogram slices of Vipp1+EPL. Tape-like spiral assemblies of Vipp1 are highlighted by dashed lines. **e**: Density profile of Vipp1 spirals and boxplot of the measured peak distances from the density profile, n = 20 (number of peaks from the same sample).

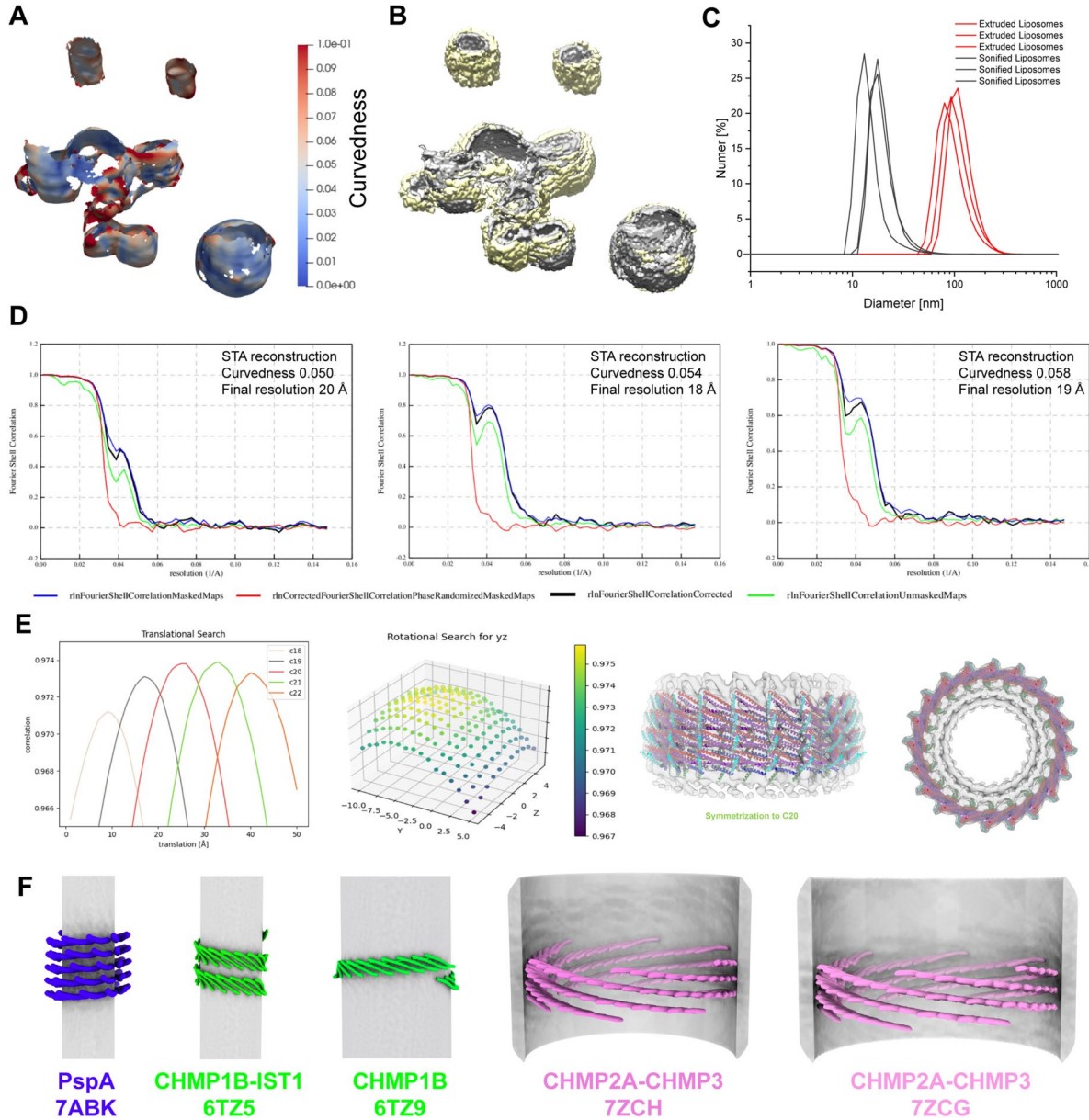

**Extended Data Fig. 9 | Vipp1 membrane and carpet segmentation, curvature estimation, resolution, and fitting of subtomogram averages. a**: Exemplary curvedness estimates conducted with surface morphometrics. **b**: Exemplary membrane (grey) and Vipp1 carpet (yellow) segmentations. **c**: Size distribution of EPL liposomes. After preparation, the size of the liposomes was checked via dynamic light scattering. Each sample was measured three times. The results are presented as a number distribution, reflecting the fraction of particles having a given diameter. The mean size ± standard error of the number distribution is 106 ± 6 nm for the extruded liposomes (red lines) and 18 ± 2 nm (black lines). This corresponds to a diameter of 143 ± 0.5 nm for the extruded liposomes, and 56 ± 0.44 nm for the sonified liposomes, based on the z-average. **d**: FSC

(0.143) for subtomogram averages with curvedness 0.05, 0.054, and 0.058, respectively. **e**: Left: correlation of translational steps with different cyclical symmetries C18, C19, C20, C21, C22 (C20 yielding the highest correlation). Right: translational and rotational search (here only plotted for y and z) over different cyclical symmetries (C18, C19, C20, C21, C22) of the 0.054 curvedness structure. The C20 symmetry gave the best correlation with the subtomogram average reconstruction. Bottom row: Symmetrized subtomogram average with rigid-body fitted and symmetrized model PDB:7O3Z. **f**: Schematic views of PspA, CHMP1B-IST1, and CHMP2A-CHMP3 α1-3 hairpins relative to the membrane axis in stacked-ring assembly, Type II, and Type I tubes.

**Extended Data Table 1 | Sample Details**

|  | Vipp1 EPL | Vipp1 (α1−6) | Vipp1 (α1−6) EPL | Vipp1 (dL10Ala) EPL |
|---|---|---|---|---|
| Protein Conc. | 4.5 mg/mL | 2 mg/mL | 2 mg/mL | 4.5 mg/mL |
| Lipid Conc. | 5 mg/mL | - | 5 mg/mL | 5 mg/mL |
| Magnification | 49 kx | 59 kx | 47 kx | 63 kx |
| Pixel size | 1.737 Å | 1.4 Å | 1.7 Å | 1.36 |
| Frames | 40 | 15 | 25 | 30 |
| Total dose | 48 e−/Å2 | 15 e−/Å2 | 25 e−/Å2 | 60 e−/Å2 |
| Defocus range | 1 to 3 µm | 1 to 2 µm | 2 to 3.5 µm | 1.25 to 2.75 µm |
| Movies | 3,217 | 5,901 | 2,463 | 29,548 |

# Reporting Summary

## Statistics

For all statistical analyses, confirm that the following items are present in the figure legend, table legend, main text, or Methods section.

| n/a | Confirmed | |
|---|---|---|
| ☐ | ☒ | The exact sample size (*n*) for each experimental group/condition, given as a discrete number and unit of measurement |
| ☐ | ☒ | A statement on whether measurements were taken from distinct samples or whether the same sample was measured repeatedly |
| ☒ | ☐ | The statistical test(s) used AND whether they are one- or two-sided<br>*Only common tests should be described solely by name; describe more complex techniques in the Methods section.* |
| ☒ | ☐ | A description of all covariates tested |
| ☒ | ☐ | A description of any assumptions or corrections, such as tests of normality and adjustment for multiple comparisons |
| ☐ | ☒ | A full description of the statistical parameters including central tendency (e.g. means) or other basic estimates (e.g. regression coefficient) AND variation (e.g. standard deviation) or associated estimates of uncertainty (e.g. confidence intervals) |
| ☒ | ☐ | For null hypothesis testing, the test statistic (e.g. *F*, *t*, *r*) with confidence intervals, effect sizes, degrees of freedom and *P* value noted<br>*Give P values as exact values whenever suitable.* |
| ☒ | ☐ | For Bayesian analysis, information on the choice of priors and Markov chain Monte Carlo settings |
| ☒ | ☐ | For hierarchical and complex designs, identification of the appropriate level for tests and full reporting of outcomes |
| ☒ | ☐ | Estimates of effect sizes (e.g. Cohen's *d*, Pearson's *r*), indicating how they were calculated |

*Our web collection on statistics for biologists contains articles on many of the points above.*

## Software and code

Policy information about availability of computer code

Data collection     Digital Micrograph Version 3.32.2403.0 ,EPU 2.12.1.2782, TIA 5.0.0.2896, FluCam viewer 6.15.3.22415, Spectra Manager(Version 2.9.0.7), Spectra Manager (Version 2.15.01)

| Data analysis | cryoSPARC v4.1 |
|---|---|
| | cryoSPARC live v4.1 |
| | ISOLDE 1.5 |
| | Coot 0.98 |
| | Phenix v1.20.1-4487) |
| | ChimeraX 1.5 |
| | OriginPro 2022b |
| | Dragonfly 2022_2 |
| | Relion 4 |
| | ctffind 4 |
| | WARP 1.1.0 beta |
| | Amira 3D 2022 |
| | AreTomo 1.1.0 |
| | Morphometrics Toolkit v0.2 |
| | ImageJ/Fiji 2.14 |
| | PyHI 8a98c25 |
| | CTFfind 4 |

For manuscripts utilizing custom algorithms or software that are central to the research but not yet described in published literature, software must be made available to editors and reviewers. We strongly encourage code deposition in a community repository (e.g. GitHub). See the Nature Portfolio guidelines for submitting code & software for further information.

# Data

Policy information about availability of data

All manuscripts must include a data availability statement. This statement should provide the following information, where applicable:
- Accession codes, unique identifiers, or web links for publicly available datasets
- A description of any restrictions on data availability
- For clinical datasets or third party data, please ensure that the statement adheres to our policy

The EMDB accession numbers for cryo-EM maps are EMD IDs:
18384, 18421, 18420, 18422, 18423, 18424, 18425, 18426, 18427, 18428, 18429, 18430, 18431, 18432, 18433, 18434, 18435, 18620, 19863, 19864, 19865, 19866, 19899, 19900, 19901, 19902, 19903, 19904
The PDB accession codes for Vipp1 models are PDB IDs:
8QFV, 8QHW, 8QHV, 8QHX, 8QHY, 8QHZ, 8QI0, 8QI1, 8QI2, 8QI3, 8QI4, 8QI5, 8QI6, 9EOM, 9EON, 9EOO, 9EOP
Available datasets accessed in this Manuscript are: PDB-ID: 7O3Y, 7O3X, 7O3W, 7O3Z

# Research involving human participants, their data, or biological material

Policy information about studies with human participants or human data. See also policy information about sex, gender (identity/presentation), and sexual orientation and race, ethnicity and racism.

| Reporting on sex and gender | N.A. |
|---|---|
| Reporting on race, ethnicity, or other socially relevant groupings | N.A. |
| Population characteristics | N.A. |
| Recruitment | N.A. |
| Ethics oversight | N.A. |

Note that full information on the approval of the study protocol must also be provided in the manuscript.

# Field-specific reporting

Please select the one below that is the best fit for your research. If you are not sure, read the appropriate sections before making your selection.

☒ Life sciences          ☐ Behavioural & social sciences          ☐ Ecological, evolutionary & environmental sciences

For a reference copy of the document with all sections, see nature.com/documents/nr-reporting-summary-flat.pdf

# Life sciences study design

All studies must disclose on these points even when the disclosure is negative.

| Sample size | Trp Fluorescence n=4, CDspectroscopy n=3, CryoEM SPA: Vipp1+EPL n=3217 micrographs, Vipp1 H1-6 n=5901 micrographs, Vipp1 H1-6 EPL |
|---|---|

| Sample size | n=2463 micrographs, Vipp1 (dL10Ala) EPL n=29548 micrographs, CryoET: 176 tilt series<br>Sample sizes were chosen based on instrument access time. The number of micrographs was sufficient to produce high resultion reconstructions of the samples. |
|---|---|
| Data exclusions | Micrographs of poor particle coverage and ice quality were discarded. |
| Replication | Due to the time-consuming nature of image acquisition and the limited access to this specialized microscope equipment, exact replicates were not performed.<br>Biochemical in vitro assay were repeated as described in the figure legends. |
| Randomization | Randomization is not applicable to the study because of the time-consuming nature of image acquisition and the limited access to this specialized microscope equipment (high-end Krios and Arctica microscopes).<br>Biochemical In vitro assays were not randomized, as this is not conventionally used in biochemical in vitro assys. |
| Blinding | Blinding experiments is not applicable to this study because of time-consuming nature of image acquisition and the limited access to this specialized microscope equipment (high-end Krios and Arctica microscopes).<br>Biochemical In vitro assays used objective quantification methods that are not susceptible to bias, so samples were not blinded. |

# Reporting for specific materials, systems and methods

We require information from authors about some types of materials, experimental systems and methods used in many studies. Here, indicate whether each material, system or method listed is relevant to your study. If you are not sure if a list item applies to your research, read the appropriate section before selecting a response.

## Materials & experimental systems

| n/a | Involved in the study |
|---|---|
| ☒ | ☐ Antibodies |
| ☒ | ☐ Eukaryotic cell lines |
| ☒ | ☐ Palaeontology and archaeology |
| ☒ | ☐ Animals and other organisms |
| ☒ | ☐ Clinical data |
| ☒ | ☐ Dual use research of concern |
| ☒ | ☐ Plants |

## Methods

| n/a | Involved in the study |
|---|---|
| ☒ | ☐ ChIP-seq |
| ☒ | ☐ Flow cytometry |
| ☒ | ☐ MRI-based neuroimaging |

## Plants

| Seed stocks | N.A. |
|---|---|
| Novel plant genotypes | N.A. |
| Authentication | N.A. |

