## [Peer Review File · Nature Structural & Molecular Biology]

Peer Review Information

Manuscript Title: Structural assemblies of membrane bound Vipp1: From loose coats and carpets to rings and rods

Corresponding author name(s): Carsten Sachse

Reviewer Comments & Decisions:

Decision Letter, initial version:

Message: 18th Dec 2023

Dear Dr. Sachse,

Thank you for submitting your manuscript "Structural assemblies of membrane bound Vipp1: From loose coats and carpets to rings and rods". The comments from the 2 reviewers who have evaluated your manuscript are below. Unfortunately, after carefully considering their comments, we cannot offer to publish your manuscript in Nature Structural & Molecular Biology.

You will see that while the referees find the work of potentially interesting, reviewer #2 in particular raises concerns about the mechanistic advance provided in the manuscript. For further consideration of the study in NSMB, we would require further development of this aspect, to provide insights into the process of Vipp1 transitioning from flat assemblies into helical structures.

If further experimentation, analysis, and revisions allow you to address the referees concerns in full, we would be prepared to consider an appeal of our decision, on the condition that no related work is published in the interim or has been accepted in our journal. Please contact me to discuss an appeal and potential revision. Please note that, until we have the opportunity to read the revised manuscript in its entirety, we cannot promise that it will be sent back for peer review.

If you were not interested in pursuing publication with NSMB on these conditions, I would be happy to consult our colleagues in Nature Communications and EMBO Press to determine their interest in this work. Please let us know how you'd like to proceed.

I am sorry we could not be more positive on this occasion. I hope that you find the referees' comments useful in deciding how best to proceed.

Sincerely,
Kat

Katarzyna Ciazynska, PhD
(she/her)
Associate Editor
Nature Structural & Molecular Biology
<https://orcid.org/0000-0002-9899-2428>

Referee expertise:

Referee #1: structural biology, cryo-EM, membrane trafficking

Referee #2: membrane remodelling, ESCRT proteins

Reviewers' Comments:

Reviewer #1:

Remarks to the Author:

Eukaryotic ESCRT-III filaments facilitate membrane remodelling for budding away from the cytosol and for repair of membrane damage. An early step in this process is the assembly of planar ESCRT-III spirals on the membrane surface. Sachse and colleagues have reconstituted the Vipp1 protein from the cyanobacterium *Synechocystis*, which is related to eukaryotic ESCRT-III subunits, on bacterial membranes. They find that the protein assembles into helical tubes, tubes formed of stacked rings, and carpets on the membrane surface. Cryo-EM of each of these types of structures shows that the Vipp1 assemblies are formed from a monomer that has a fold like ESCRT-III subunits. Each of these assemblies formed from intact Vipp1 has the protein arranged in helical or ring-like arrays on the surface of a lipid vesicle or tubule. They have shown that helix α_0 forms the prominent feature interacting with membranes and this helix appears to partially enter the outer leaflet of the lipid bilayer. This form of membrane interaction makes the Vipp1 assemblies very sensitive to membrane curvature and gives Vipp1 the ability to tubulate lipid membranes and cause membrane budding from vesicles. critical for the membrane interaction. The authors have generated a wide range of structures from the many types of vesicles and tubes that they see in the samples. These structures provide a clear mechanistic basis for how the assemblies can adjust to a range of membrane curvatures. One interesting conclusion that they make is that Vipp1 assemblies have greatest order, best binding and most homogeneity on lipid vesicles or tubules with high curvature. On low curvature membranes, the assemblies are flatter and much more heterogeneous. This observation suggests that on completely flat membranes of supported lipid bilayers Vipp1 might be expected to form much less ordered assemblies.

The manuscript makes great contribution in several respects:

1. It provides a resolution of Vipp1 assemblies that is sufficient to understand the structure and to understand how hinges within the protein bend to accommodate a range

of lipid tubes and vesicles.

2. By classifying the many different types of vesicles and Vipp1 assemblies present in their samples, the authors have been able to establish a plausible dynamic view of how the protein can bind to a membrane, induce membrane tubulation and break membrane tubes into smaller vesicles. These inferred dynamics should be invaluable for understanding the mechanistic basis of the membrane sculpting abilities of ESCRT-III subunits.

3. The authors have unequivocally shown that helix alpha0 is critical for binding to membranes and for inducing and sensing membrane curvature. In the absence of alpha0, Vipp1 assembles, but it does not assemble around lipid tubes.

4. The stacked ring assemblies suggest a mechanism for Vipp1 to break membrane tubes into Vipp1-clad vesicles.

5. It provides a very different view of Vipp1 assemblies than would be possible on flat membranes, using supported lipid bilayers. This is especially valuable because it is likely to be more generally applicable to assemblies in cells.

The authors have characterised cryo-EM helical and ring structures assembled on lipid tubes. The cryo-EM and associated model building appears to be of a high standard. They provide moderate resolution reconstructions that are sufficient for the conclusions presented in the text. These observations go beyond the previously reported Vipp1 rings, and they suggest the conformational transitions from flat patches through tubes to segmented tubes to smaller vesicles. These structures show clear relationships with eukaryotic ESCRT-III assemblies. These transitions are central to the many roles that ESCRT-III family subunits have in sculpting and repairing membranes.

The manuscript is very well written, and I have not spotted any problem that needs to be addressed. I believe it should be accepted in its present form.

Reviewer #2:

Remarks to the Author:

A) Summary of key results:

ESCRT-III family proteins comprise the critical machinery that drive membrane remodeling and fission processes in fundamental cellular processes such as cytokinesis, nuclear membrane repair, endosomal repair and multivesicular body formation. Cryo-EM structures of the bacterial proteins Vipp1 and PspA, reported in a series of very nice Cell papers from 2021 (PMCID: PMC8281802, PMID: 34166613 and PMID: 34166616), revealed that bacteria also contain ESCRT-III assemblies (mainly ring-like structures), a finding that was previously unknown. The authors of the current manuscript described in their 2021 Cell paper (PMID: 34166616) how PspA contains an ESCRT-III assembly and reported a medium-resolution (7 Angstrom) reconstruction of membrane-bound PspA but could not unambiguously assign membrane-protein interfaces. The two other 2021 Cell papers that reported Vipp1 structures did not contain membrane-bound reconstructions of Vipp1.

The authors of the current work have now switched their focus from PspA to Vipp1 and can report on the following:

1) Membrane bound Vipp1 assemblies (and report medium-resolution reconstructions of these assemblies)

- 2) Membrane-bound Vipp1 assemblies can form helical structures. Previously, only ring-like structures were reported for Vipp1 assemblies (whereas helical assemblies for yeast and mammalian ESCRT-III proteins have been described in the past. The authors describe 3 helical reconstructions.
- 3) The authors report 3 stacked-ring structures encasing membrane tubes
- 4) The authors report 3 structures (using subtomogram averaging techniques) of membrane-bound 'carpets'
- 4) Helix-0 of Vipp1 is clearly the membrane interacting domain of Vipp1 assemblies and the authors report an additional 10 structures of Vipp1 with its helix-0 truncated

B) Originality and significance:

Originality:

Membrane bound Vipp1 assembly reconstructions have not previously been reported. The use of a denaturing and refolding-in-the-presence-of-bacterial-membranes strategy (used by the authors for PspA in PMID: 34166616) is clever and was the technical advance that allowed the authors to visualize different forms of membrane-bound Vipp1 assemblies, namely flat 'carpets' stacked-rings and helical rods.

The fact that Vipp1 helical assemblies appear to change their orientation with respect to the plane of the membrane tube is intriguing.

Significance:

The major outstanding question in the ESCRT field is how ESCRT-III assemblies can mechanistically transition from flat spirals to helices to drive membrane fission. The fact that the bacterial ESCRT-III proteins (in this case represented by Vipp1) can form helical assemblies means that this system can be used to explain this transition. Such a mechanistic understanding would be a major breakthrough in the field of ESCRT-III biology and of interest in all membrane fission disciplines.

This paper attempts to describe this transition. The limitations of the current work are that the findings are somewhat observational don't explain in mechanistic detail how the transition from flat spirals (carpets) to either rings or helical assemblies occurs. The authors are somewhat unlucky that their medium-resolution cryo-EM reconstructions limit their ability somewhat to make detailed mechanistic claims.

Data and Methodology:

The cryoEM approaches to solve the structures of the various membrane-bound Vipp1 assemblies are technically appropriate. Using a denaturing-refolding approach to generate the various membrane-bound Vipp1 assemblies is clever and well executed. The quality of the data is sound but the medium resolution of the cryo-EM reconstructions (5-7 Angstroms for the helical and stacked-ring reconstructions) somewhat limits the overall quality to the data. Equivalent mammalian membrane-bound ESCRT-III reconstructions (PMCID: PMC7343221 and PMID: 36604498) in the mid-3 Angstrom resolution allowed those studies to make more mechanistic claims.

The quality of the data presentation is good.

Suggested improvements:

If the authors were able to push the resolution of some of their reconstructions, especially to reveal the details of how helix-0 interacts with the outer leaflet of the membrane, this would be a significant improvement to the results.

~55,000 segments (particles) resulting in a reconstruction of 5 Angstroms for the Type I helical tubes seems comparable to other ESCRT-III reconstructions. (and higher particle numbers have been required to obtain higher resolutions in some cases). Have the authors attempted to collect additional data on this sample to see if they can push the resolution of this particular reconstruction? This sample appears to be the most promising for the critical jump in resolution that would elevate this paper

Does the resolution of the helical reconstructions allow the authors to make and test mutations that might lock (or favor) any of the different orientations seen in the stacked ring assemblies and the Type I and Type II helical assemblies?

References:

The authors have referenced previous work in the field appropriately

Clarity and content:

The authors have provided appropriate context for the importance of the. Some of the language used throughout the manuscript could be improved to provide greater clarity (see specific comments below for some suggestions)

Specific comments:

Line 51: The authors might consider including the very recent review on ESCRT-III proteins across the tree of life from Carlton and Baum (PMID: 37944425) as part of this series of references.

Line 64: The authors might consider adding their nice 2023 TiBS review article (Schlosser et al) to the list of papers describing previously published Vipp1 structures.

Line 114: The negative stained images in Supp. Fig. 1A do not contain sufficient resolution to support the claim that these assemblies are rings and short stacks of rings. Even when zooming in on the digital version of these panels, one can only see that there are filamentous structures present - but it cannot be claimed that these are rings or stacked rings. This claim is probably best suited to referencing Figure 1A (bottom panels)

Line 126: Fig. 1B Bottom (the 2D classes) does not prove that what a reader is looking at are stacked rings, helical tubes and flat carpets. That can only be claimed from the actual reconstructions in later figures.

Line 130: the authors claim that the 2D class average clearly show membrane density in the stacked-ring assemblies. At this resolution, it is really not feasible to make this claim. can the authors point out this density in these panels? Figure 3 is the more appropriate

place to make this claim.

Line 131: the authors claim that the 2D class average clearly show membrane density in both types of helical tubes. At this resolution, it is really not feasible to make this claim. Can the authors point out this density in these panels? Figure 2D is the more appropriate place to make this claim.

Line 134: Can the authors recheck the scale bar on the Type 1 tube 2D class averages. The claim is the most of these tubes are 300 Angstroms in diameter, but from the scale bar in the images, the diameter appears to be just over 200 Angstroms.

Line 136: Similar to the above comment; these tubes do not appear to be 270-290 Angstroms in diameter, based on the provided scale bar.

Line 139: Can the authors point out the pronounced spikes on the 2D class averages they report in the text. It may also help the reader if all the carpet 2D class averages were center aligned.

Line 152-Line157; Where is the data supporting this description? I suggest referring to the relevant Figure/panel each time a new claim is made.

Line 158: Figure 2A. I suggest that to help the flow of this Figure, that the authors start with a description of panel D first and then move to the higher-resolution details in B and C after. (Follow the layout of Figure 3).

Line 181: In addition to the green arrow pointing out the indentation into figure, I suggest also labelling helix 0 in the actual figure (Figure 2F)

Line 206-207: Refer to the exact figures that each of these claims refer to.

Line 210: Refer to the exact figure here that shows the C11 and C14 stacked rings

Line 232-234. Refer to the exact Figure here for this claim

Line 257: I really is not possible to tell from the 2D class averages in Suppl. Fig. 4A at this resolution that these structures are helical in nature and not stacked rings. that claim can be made in Suppl. Fig. 4C.

Line 264: The 2D class averages in Suppl. Fig 4A and 4B do not have scale bars that would allow is to judge the claim that the diameters are 280 to 420 Angstroms in diameter.

Line 289-290: Refer to the Figure here that supports this claim.

Line 302-303: Can the authors clarify what the distances are that they are referring to (i.e distances between what and what?)

Line 323: Which Figure supports the claim in this sentence? - refer to it here.

Line 345: Refer to the figure/panel here that shows the membrane-attached carpets

Line 438-429: Which Figure support this claim?

Line 436-438: which figure does this description refer to?

Line 460-463: A comparison figure (Supplementary) would be very helpful here to help the reader understand the differences in the orientations relative to the membrane axes.

Line 473-474: Can the authors clarify their claim here. Are they attempting to say that the angle/orientation of how helix 0 is oriented with respect to the membrane is different for Vipp1 compared to PspA? CHMP1B does not have a helix 0 so it is hard to make that exact comparison.

Figure 7B: It would be very helpful to a reader to label each panel something like i), ii), iii) etc to help follow which image/cartoon is being described at each proposed stage.

Author Rebuttal to Initial comments

NSMB-A48465

We thank the reviewers for carefully reading the manuscript and for their positive overall feedback. Below we provide a detailed point-by-point response to address the raised questions. In a revised manuscript, we added a high-resolution 3.0 Å structure of a stabilizing Hinge 2 mutant that showed significantly reduced conformational and structural plasticity. The structural details of helix $\alpha 0$ were now revealed in full and allow mechanistic conclusions on the mode of membrane interaction. The revised manuscript has all additions highlighted in yellow while stretches that were moved within the manuscript have a green highlighted color.

Reviewer #1:

The manuscript makes great contribution in several respects:

1. It provides a resolution of Vipp1 assemblies that is sufficient to understand the structure and to understand how hinges within the protein bend to accommodate a range of lipid tubes and vesicles.
2. By classifying the many different types of vesicles and Vipp1 assemblies present in their samples, the authors have been able to establish a plausible dynamic view of how the protein can bind to a membrane, induce membrane tubulation and break membrane tubes into smaller vesicles. These inferred dynamics should be invaluable for understanding the mechanistic basis of the membrane sculpting abilities of ESCRT-III subunits.
3. The authors have unequivocally shown that helix $\alpha 0$ is critical for binding to membranes and for inducing and sensing membrane curvature. In the absence of $\alpha 0$, Vipp1 assembles, but it does not assemble around lipid tubes.
4. The stacked ring assemblies suggest a mechanism for Vipp1 to break membrane tubes into Vipp1-clated vesicles.
5. It provides a very different view of Vipp1 assemblies than would be possible on flat membranes, using supported lipid bilayers. This is especially valuable because it is likely to be more generally applicable to assemblies in cells.

The authors have characterised cryo-EM helical and ring structures assembled on lipid tubes. The cryo-EM and associated model building appears to be of a high standard. They provide moderate resolution reconstructions that are sufficient for the conclusions presented in the text. These observations go beyond the previously reported Vipp1 rings, and they suggest the conformational transitions from flat patches through tubes to segmented tubes to smaller vesicles. These structures show clear relationships with eukaryotic ESCRT-III assemblies. These transitions are central to the many roles that ESCRT-III family subunits have in sculpting and repairing membranes.

The manuscript is very well written, and I have not spotted any problem that needs to be addressed. I believe it should be accepted in its present form.

We thank reviewer #1 for the very positive assessment of our manuscript. The referee somewhat critically remarks that the presented Vipp1 structures were determined at moderate resolutions (5-8 Å). In agreement with this assessment, we were very careful in limiting our interpretations to the resolution. As we received a similar comment as the main item from Referee #2, we generated two 3.0 Å resolution maps that we generated from a particular plasticity-restrained mutant located in Hinge 2. Given the limited resolution presented in our first version of the manuscript, the interesting interactions of helix $\alpha 0$ with the membrane were not detailed. In the revised version of the manuscript (follow new sections pages 11- 15, new Figure 4), we now also obtained a detailed picture of the involved residues and how they are positioned to the membrane. Moreover, we added functional biophysical data on the isolated helix $\alpha 0$ peptide including residue mutants to characterize the peptide membrane interaction in detail. As both of these items were also raised by referee #2, we refer the reader to the detailed given below.

Reviewer #2:

The limitations of the current work are that the findings are somewhat observational don't explain in mechanistic detail how the transition from flat spirals (carpets) to either rings or helical assemblies occurs. The authors are somewhat unlucky that their medium-resolution cryo-EM reconstructions limit their ability somewhat to make detailed mechanistic claims.

Suggested improvements:

If the authors were able to push the resolution of some of their reconstructions, especially to reveal the details of how helix-0 interacts with the outer leaflet of the membrane, this would be a significant improvement to the results.

~55,000 segments (particles) resulting in a reconstruction of 5 Angstroms for the Type I helical tubes seems comparable to other ESCRT-III reconstructions. (and higher particle numbers have been required to obtain higher resolutions in some cases). Have the authors attempted to collect additional data on this sample to see if they can push the resolution of this particular reconstruction? This sample appears to be the most promising for the critical jump in resolution that would elevate this paper.

Does the resolution of the helical reconstructions allow the authors to make and test mutations that might lock (or favor) any of the different orientations seen in the stacked ring assemblies and the Type I and Type II helical assemblies?

Instigated by the comment of the reviewer #2, we designed a Hinge 2 mutant that reduced the observed structural plasticity of Vipp1 assemblies. With the help of this mutation, we show that this hinge is critical for the structural diversity we describe in the manuscript, and thus, now identified the mechanism of Vipp1 structural adjustments. Furthermore, we were now able to solve the structure of Type II tubes at 3.0 Å resolution and revealed the detailed conformation of the helix α_0 as well as critical conserved interactions. We now describe the new results on page 12-15 and in the new Figure 4 of the revised version of the manuscript. In addition to the new high-resolution structural details of α_0 , we characterized the contribution of particular residues in isolated α_0 helix peptides using biochemical membrane interaction assays and included the new data in the results section (page 8):

To experimentally test which helix α_0 residues are involved in membrane interaction, we made use of the observation that the isolated Vipp1 helix α_0 forms an α -helix only when binding to negatively charged membrane surfaces (McDonald et al., 2017). In this way, we measured the membrane binding capacity of different variants of the helix α_0 peptide using CD spectroscopy. Compared with the WT peptide, peptides that had R6 and R9 or F4 and V11 replaced by two alanines reduced the membrane affinity to a small extent as they reached similar ellipticity plateaus, while the WT peptide binding curve decayed faster than the R6AR9A and F4AV11A binding curves (Suppl. Fig. 7I). By replacing the eight hydrophobic residues with alanines, thus destroying the amphipathic character of the peptide, we abolished membrane binding completely. This observation indicates that the amphipathic character is essential for membrane interaction while the hydrophobic face appears to be more critical than the positively charged residues.

In summary, prompted by the reviewer's comments and suggestions, we now identified Hinge 2 being key for the observed structural plasticity of Vipp1, and using the Hinge 2 mutant with reduced flexibility, we were able to solve the structure of a Vipp1 assembly at near atomic resolution. Using this high-resolution structure, we could now identify α_0 residues that connect different helices of adjacent subunits to stabilize Vipp1 type assemblies, and we characterized α_0 residues that are involved in lipid binding.

Specific comments:

1. Line 51: The authors might consider including the very recent review on ESCRT-III proteins across the tree of life from Carlton and Baum (PMID: 37944425) as part of this series of references.

Added as suggested.

2. Line 64: The authors might consider adding their nice 2023 TiBS review article (Schlosser et al) to the list of papers describing previously published Vipp1 structures.

Added as suggested.

3. Line 114: The negative stained images in Supp. Fig. 1A do not contain sufficient resolution to support the claim that these assemblies are rings and short stacks of rings. Even when zooming in on the digital version of these panels, one can only see that the that are filamentous structures present - but it cannot be claimed that these are rings or stacked rings. This claim is probably best suited to referencing Figure 1A (bottom panels)

In order to support the claim, we replaced old micrographs with new micrographs at higher resolution and containing more clearly visible rings in Supp. Fig. 1.

4. Line 126: Fig. 1B Bottom (the 2D classes) does not prove that what a reader is looking at are stacked rings, helical tubes and flat carpets. That can only be claimed from the actual reconstructions in later figures.

In order to clarify the statement, we restated the sentences:

We scrutinized the Vipp1 cryo-EM micrographs and identified isolated ring complexes as well as elongated rods of stacked rings, tubes, and vesicles covered with carpet structures (Fig. 1B top). The observed structures are heterogeneous, *e.g.*, ring complexes have different sizes and the tubular structures are curved, have kinks and variable diameters along the tube axis. When we analyzed the segmented Vipp1 polymer structures by image classification methods, we found 48 % being present as elongated evenly indented rods of stacked rings, 45 % consisting of two types of tubes with distinct helical lattices and 7 % corresponding to regular two-dimensional carpet structures (Fig. 1B bottom).

We respectfully disagree with the reviewer's assessment that the architectural claims can only be stated from the 3D reconstructions. The 2D class averages reveal characteristic helical as well as the 2D lattice features of the Vipp1 assemblies while the indented rods are only compatible with stacked rings of rotational symmetry.

5. Line 130: the authors claim that the 2D class average clearly show membrane density in the stacked-ring assemblies. At this resolution, it is really not feasible to make this claim. can the authors point out this density in these panels? Figure 3 is the more appropriate place to make this claim.

We removed the statement as Figure 3 unambiguously shows the membrane density.

6. Line 131: the authors claim that the 2D class average clearly show membrane density in both types of helical tubes. At this resolution, it is really not feasible to make this claim. Can the authors point out this density in these panels? Figure 2D is the more appropriate place to make this claim.

We removed the statement as Figure 2B/C unambiguously shows the membrane density.

7. Line 134: Can the authors recheck the scale bar on the Type 1 tube 2D class averages. The claim is the most of these tubes are 300 Angstroms in diameter, but from the scale bar in the images, the diameter appears to be just over 200 Angstroms.

Based on the comment, we re-checked the scale bar. We included the extended spike density in the diameter measurement and therefore arrive at 300 Å.

8. Line 136: Similar to the above comment; these tubes do not appear to be 270-290 Angstroms in diameter, based on the provided scale bar.

Based on the comment, we re-checked the scale bar. We measured an apparent outer diameter of 250 – 260 Å slightly lower than in the cross-section of the 3D structure. We have corrected the text accordingly.

9. Line 139: Can the authors point out the pronounced spikes on the 2D class averages they report in the text. It may also help the reader if all the carpet 2D class averages were center aligned.

To improve the visualization of our results, we now aligned the 2D classes and updated the Figure 1B bottom. We also added purple arrowheads indicating the spikes in Figure 1B top and Figure 1B bottom.

10. Line 152-Line157; Where is the data supporting this description? I suggest referring to the relevant Figure/panel each time a new claim is made.

As suggested, we moved the statement referring to the figure up. The revised text reads:

Subsequently, we built atomic models of the tubes by flexibly fitting a previously solved monomer structure (PDB:7O3W) into the determined cryo-EM density maps. The Vipp1 secondary structure consists of six α -helices connected by short loops plus a predicted 7th α -helix at the C-terminus (Fig. 2A shows the ESCRT-III unified nomenclature for the Vipp1 helices (Schlösser et al., 2023)).

11. Line 158: Figure 2A. I suggest that to help the flow of this Figure, that the authors start with a description of panel D first and then move to the higher-resolution details in B and C after. (Follow the layout of Figure 3).

As suggested, we rearranged Figure 2 and the corresponding text accordingly to improve the flow and follow layout of Figure 3.

12. Line 181: In addition to the green arrow pointing out the indentation into figure, I suggest also labelling helix 0 in the actual figure (Figure 2F)

As suggested, we added the helix labels to the Figure 2F.

13. Line 206-207: Refer to the exact figures that each of these claims refer to.

According to editorial guidelines, one should avoid the multiple redundant referrals to the same Figure. Nevertheless, due to the central role of this micrograph in the manuscript, we made an exception to this rule and added a re-occurring reference: (see Fig. 1B).

14. Line 210: Refer to the exact figure here that shows the C11 and C14 stacked rings.

We added the reference of the data earlier, as requested. The revised sentence reads:

Based on the cryo-EM images, we determined the 3D structures of C11 to C14 stacked ring assemblies at resolutions of 6.8 to 6.9 Å (Fig. 3A, Table 1).

15. Line 232-234. Refer to the exact Figure here for this claim

We moved the Figure reference to the mentioned sentence:

In contrast to the thickness of the lipid bilayer engulfed by helical Vipp1 tubes, the bilayer thickness in stacked Vipp1 rings was generally smaller and more variable (Suppl. Fig. 3D).

16. Line 257: I really is not possible to tell from the 2D class averages in Suppl. Fig. 4A at this resolution that these structures are helical in nature and not stacked rings. that claim can be made in Suppl. Fig. 4C.

We agree with the reviewer and, therefore, removed “helical” as requested.

17. Line 264: The 2D class averages in Suppl. Fig 4A and 4B do not have scale bars that would allow is to judge the claim that the diameters are 280 to 420 Angstroms in diameter.

We added scale bars as requested.

18. Line 289-290: Refer to the Figure here that supports this claim.

According to editorial guidelines, one should avoid the multiple redundant referrals to the same Figure. Nevertheless, due to the central role of this micrograph in the manuscript, we made an exception and added a re-occurring reference: (see Fig. 1B).

19. Line 302-303: Can the authors clarify what the distances are that they are referring to (i.e distances between what and what?)

We clarified the statement in the text, which is clearly supported by the measured distances in Supp Fig. 7 A and B:

A comparison of the cylindrical projection of the Vipp1 Type I tubes and 2D class averages of the carpet and their corresponding Fourier transforms revealed that the distances occurring in this regular stripe and spike pattern were very similar, indicating common underlying structures (Suppl. Fig. 8A and B). These distances of the cylindrical projection in Type I tubes correspond to the stacking distance of the $\alpha 1/\alpha 2$ helical hairpin array suggesting that a similar hairpin array is also present in Vipp1 carpets.

20. Line 323: Which Figure supports the claim in this sentence? - refer to it here.

As the relevant figures were referenced one or two sentences above, we chose not to violate editorial guidelines of avoiding multiple redundant figure referrals.

21. Line 345: Refer to the figure/panel here that shows the membrane-attached carpets

It is an introductory sentence of the paragraph summarizing previously described carpet structures. Thus, we prefer to stick to the editorial guidelines of avoiding multiple redundant figure referrals. Nevertheless, when further requested and allowed by the editors, we can here again refer to the figure.

22. Line 428-429: Which Figure support this claim?

To help the reader, we added a re-occurring figure referral (see Fig. 3A).

23. Line 436-438: which figure does this description refer to?

To help the reader, we added a re-occurring figure referral (see Fig. 3C and D).

24. Line 460-463: A comparison figure (Supplementary) would be very helpful here to help the reader understand the differences in the orientations relative to the membrane axes.

As suggested, we now added Suppl. Fig. 8F for this comparison.

25. Line 473-474: Can the authors clarify their claim here. Are they attempting to say that the angle/orientation of how helix 0 is oriented with respect to the membrane is different for Vipp1 compared to PspA? CHMP1B does not have a helix 0 so it is hard to make that exact comparison.

We apologize for the incorrect helix assignment and amended the statement:

Similar to other ESCRT-III proteins, the membrane contact in the Vipp1 tubes and rings is mediated either by an additional α -helix (*i.e.* $\alpha 0$) in the case of Vipp1 and *SynPspA* or a short unstructured extension of helix $\alpha 1$ in the case of CHM1B or CHMP2A+CHMP3 (Azad et al., 2023; Junglas et al., 2021; Moss et al., 2023; Nguyen et al., 2020). However, in contrast to *SynPspA* helix $\alpha 0$ that protrudes towards the bilayer (Junglas et al., 2021), Vipp1 helix $\alpha 0$ is lying flat in the membrane plane and appears to be partially submerged into the outer bilayer leaflet.

26. Figure 7B: It would be very helpful to a reader to label each panel something like i), ii), iii) etc to help follow which image/cartoon is being described at each proposed stage.

As suggested, we added the requested labels to the Figure 7B. We agree that this helps the reader to follow our model.

Decision Letter, first revision:

Message: Our ref: NSMB-A48465A-Z

22nd May 2024

Dear Dr. Sachse,

Thank you for submitting your revised manuscript "Structural assemblies of membrane bound Vipp1: From loose coats and carpets to rings and rods" (NSMB-A48465A-Z). It has now been seen by the original referees and their comments are below. The reviewers find that the paper has improved in revision, and therefore we'll be happy in principle to publish it in Nature Structural & Molecular Biology, pending minor revisions to satisfy the referees' final requests and to comply with our editorial and formatting guidelines.

To facilitate our work at this stage, it is important that we have a copy of the main text as a word file. If you could please send along a word version of this file as soon as possible, we would greatly appreciate it; please make sure to copy the NSMB account (cc'ed above).

Sincerely,

Katarzyna Ciazynska, PhD
(she/her)
Associate Editor
Nature Structural & Molecular Biology
<https://orcid.org/0000-0002-9899-2428>

Reviewer #2 (Remarks to the Author):

In this revised manuscript, the authors have made a significant advance on the original submission. Namely, they were able to form Vipp1 filaments with reduced conformational and structural plasticity, by designing a stabilizing hinge 2 mutant. A subset of these filaments were solved to 3 Angstroms resolution (compared to the medium resolution of filaments in the original submission). This has now allowed the authors to very nicely describe in molecular detail the interactions of helix0 with i) the membrane and ii) other intra-filament helices.

The main limitation of the original submission was that the findings were somewhat

observation. The level of detail now provided by the 3 Angstrom resolution structures significantly strengthens this paper and allows the authors to clearly draw mechanistic conclusions on the mode of membrane interaction.

This revision has answered/met all my original concerns and should be accepted immediately for publication. I was particularly impressed that the authors took the time and effort to generate the more 'regular' filaments, thereby allowing them to push the resolution. The result is a really nice piece of work.

Final Decision Letter:

Message: 5th Sep 2024

Dear Dr. Sachse,

We are now happy to accept your revised paper "Structural assemblies of membrane bound Vipp1: From loose coats and carpets to rings and rods" for publication as an Article in Nature Structural & Molecular Biology.

Due to the importance of these deadlines, we ask that you please let us know now whether you will be difficult to contact over the next month. If this is the case, we ask you provide us with the contact information (email, phone and fax) of someone who will be able to check

the proofs on your behalf, and who will be available to address any last-minute problems.

Your paper will be published online soon after we receive proof corrections and will appear in print in the next available issue. You can find out your date of online publication by contacting the production team shortly after sending your proof corrections.

If you have not already done so, we strongly recommend that you upload the step-by-step protocols used in this manuscript to the Protocol Exchange. Protocol Exchange is an open online resource that allows researchers to share their detailed experimental know-how. All uploaded protocols are made freely available, assigned DOIs for ease of citation and fully searchable through nature.com. Protocols can be linked to any publications in which they are used and will be linked to from your article. You can also establish a dedicated page to collect all your lab Protocols. By uploading your Protocols to Protocol Exchange, you are enabling

researchers to more readily reproduce or adapt the methodology you use, as well as increasing the visibility of your protocols and papers. Upload your Protocols at www.nature.com/protocolexchange/. Further information can be found at www.nature.com/protocolexchange/about.

Please note that *Nature Structural & Molecular Biology* is a Transformative Journal (TJ). Authors may publish their research with us through the traditional subscription access route or make their paper immediately open access through payment of an article-processing charge (APC). Authors will not be required to make a final decision about access to their article until it has been accepted. Find out more about Transformative Journals

Sincerely,

Katarzyna Ciazynska, PhD
(she/her)
Associate Editor
Nature Structural & Molecular Biology
<https://orcid.org/0000-0002-9899-2428>